# Increased theta/alpha synchrony in the habenula-prefrontal network with negative emotional stimuli in human patients

Yongzhi Huang[1,2†], Bomin Sun[3†], Jean Debarros[4], Chao Zhang[3], Shikun Zhan[3], Dianyou Li[3], Chencheng Zhang[3], Tao Wang[3], Peng Huang[3], Yijie Lai[3], Peter Brown[4], Chunyan Cao[3]*, Huiling Tan[4]*

[1]Academy of Medical Engineering and Translational Medicine, Tianjin University, Tianjin, China; [2]Nuffield Department of Surgical Sciences, University of Oxford, Oxford, United Kingdom; [3]Department of Neurosurgery, Affiliated Ruijin Hospital, Shanghai Jiao Tong University School of Medicine, Shanghai, China; [4]Medical Research Council (MRC) Brain Network Dynamics Unit at the University of Oxford, Nuffield Department of Clinical Neurosciences, University of Oxford, Oxford, United Kingdom

**Abstract** Lateral habenula is believed to encode negative motivational stimuli and plays key roles in the pathophysiology of psychiatric disorders. However, how habenula activities are modulated during the processing of emotional information is still poorly understood. We recorded local field potentials from bilateral habenula areas with simultaneous cortical magnetoencephalography in nine patients with psychiatric disorders during an emotional picture-viewing task. Transient activity in the theta/alpha band (5–10 Hz) within the habenula and prefrontal cortical regions, as well as the coupling between these structures, is increased during the perception and processing of negative emotional stimuli compared to positive emotional stimuli. The increase in theta/alpha band synchronization in the frontal cortex-habenula network correlated with the emotional valence but not the arousal score of the stimuli. These results provide direct evidence for increased theta/alpha synchrony within the habenula area and prefrontal cortex-habenula network in the perception of negative emotion in human participants.

*For correspondence:
ccy40646@rjh.com.cn (CC);
huiling.tan@ndcn.ox.ac.uk (HT)

†These authors contributed equally to this work

Competing interests: The authors declare that no competing interests exist.

## Introduction

The habenula is an epithalamic structure that functionally links the forebrain with the midbrain structures that are involved in the release of dopamine (i.e., the substantia nigra pars compacta and the ventral tegmental area) and serotonin (i.e., raphe nucleus) (*Wang and Aghajanian, 1977*; *Herkenham and Nauta, 1979*; *Hikosaka et al., 2008*; *Hong et al., 2011*; *Proulx et al., 2014*; *Hu et al., 2020*). As a region that could influence both the dopaminergic and serotonergic systems, the habenula is thought to play a key role in not only sleep and wakefulness but also in regulating various emotional and cognitive functions. Animal studies showed that activities in lateral habenula (LHb) increased during the processing of aversive events such as omission of predicted rewards, and stimuli provoking anxiety, stress, pain and fear (*Matsumoto and Hikosaka, 2007*; *Hikosaka, 2010*; *Yamaguchi et al., 2013*; *Hu et al., 2020*).

Hyperexcitability and dysfunction of the LHb have been implicated in the development of psychiatric disorders including depressive disorder and bipolar disorders (*Fakhoury, 2017*; *Yang et al., 2018b*). In rodents, LHb firing rate and metabolism is elevated in parallel with depressive-like

phenotypes such as reduction in locomotor and rearing behaviors (*Caldecott-Hazard et al., 1988*). Habenula activities also increase during acquisition and recall of conditioned fear (*González-Pardo et al., 2012*). In human participants, high-resolution magnetic resonance imaging has revealed smaller habenula volume in patients with depressive and bipolar disorders (*Savitz et al., 2011a*). Dysfunction of the LHb has also been involved in different cognitive disorders, such as schizophrenia (*Shepard et al., 2006*) and addiction (*Velasquez et al., 2014*). More direct evidence of the involvement of the LHb in psychiatric disorders in humans comes from deep brain stimulation (DBS) of the LHb that has potential therapeutic effects in treatment-resistant depression, bipolar disorder, and schizophrenia (*Sartorius et al., 2010*; *Zhang et al., 2019*; *Wang et al., 2020*). However, how habenula activities are modulated during the processing of emotional information in humans is still poorly understood.

The processing of emotional information is crucial for an individual's mental health and has a substantial influence on social interactions and different cognitive processes. Dysfunction and dysregulation of emotion-related brain circuits may precipitate mood disorders (*Phillips et al., 2003b*). Investigating the neural activities in response to emotional stimuli in the cortical-habenula network is crucial to our understanding of emotional information processing in the brain. This might also shed light on how to modulate habenula in the treatment of psychiatric disorders. In this study, we utilize the unique opportunity offered by DBS surgery targeting habenula as a potential treatment for psychiatric disorders. We measured local field potentials (LFPs) from the habenula area using the electrodes implanted for DBS in patients during a passive emotional picture-viewing task (*Figure 1*; Materials and methods). Whole-brain magnetoencephalography (MEG) was simultaneously recorded. This allowed us to investigate changes in the habenula neural activity and its functional connectivity with cortical areas induced by the stimuli of different emotional valence. The high temporal resolution of the LFP and MEG measurements also allowed us to evaluate how local activities and cross-region connectivity change over time in the processing of emotional stimuli. Previous studies on rodent models of depression showed that, during the depression-like state in rodents, LHb neuron firing increased with the mean firing rate at the theta band (*Li et al., 2011*) and LHb neurons fire in bursts and phase locked to local theta band field potentials (*Yang et al., 2018a*). Therefore, we hypothesize that theta band activity in the habenula LFPs in humans would increase in response to negative emotional stimuli.

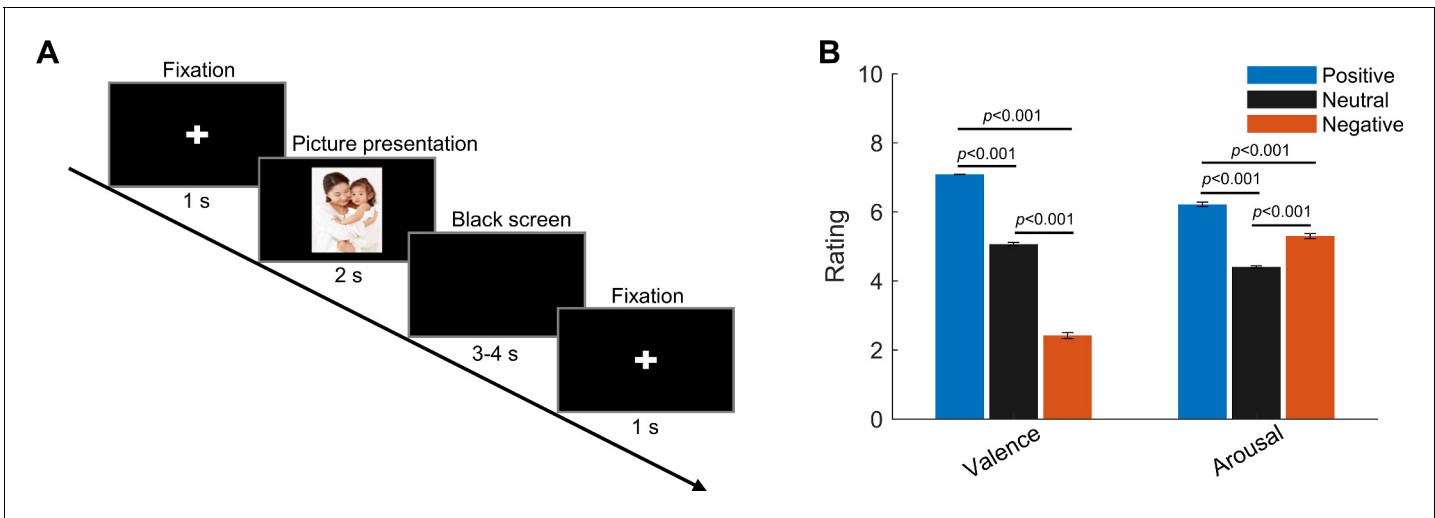

**Figure 1.** Experimental paradigm and ratings (valence and arousal) of the presented pictures. (**A**) Timeline of one individual trial: each trial started with a white cross ('+') presented with black background for 1 s, indicating the participants to get ready and pay attention; then a picture was presented in the center of the screen for 2 s. This was followed by a blank black screen presented for 3–4 s (randomized). (**B**) Valence and arousal ratings for figures of the three emotional categories presented to the participants. Valence: 1 = very negative; 9 = very positive; arousal: 1 = very clam; 9 = very exciting. Error bars indicate the standard deviation of the corresponding mean across participants (N = 9).

## Results

### Spontaneous oscillatory activity in the habenula during rest includes theta/alpha oscillations

Electrode trajectories and contact positions of all recorded patients in this study were reconstructed using the Lead-DBS toolbox (*Horn and Kühn, 2015*) and shown in *Figure 2A*. The peak frequency of the oscillatory activities during rest for each electrode identified using the Fitting Oscillations and One-Over-F (FOOOF) algorithm (*Haller et al., 2018*; *Donoghue et al., 2020*) is presented in *Table 1*. We detected the power of oscillatory activities peaking in the theta/alpha frequency range (here defined as 5–10 Hz) in 13 out of the 18 recorded habenula during rest compared to 7 of the 18 recorded habenula with peaks in beta band (12–30 Hz) (*Figure 2B*). The average peak frequency was 8.2 ± 1.1 Hz (ranges from 6.1 Hz to 10 Hz) for theta/alpha, and 15.1 ± 1.8 Hz (ranges from 12.5 Hz to 16.9 Hz) for beta band (*Figure 2C*). 3 out of the 18 recorded habenula showed oscillatory activities in both theta/alpha and beta bands. *Figure 2D–F* shows the position of the electrodes with only theta/alpha band peaks, with only beta peaks in both sides (Case 3), with both theta/alpha and beta band peaks during rest (Case 6), respectively. The electrodes from which only alpha/theta peaks were detected are well placed in the habenula area.

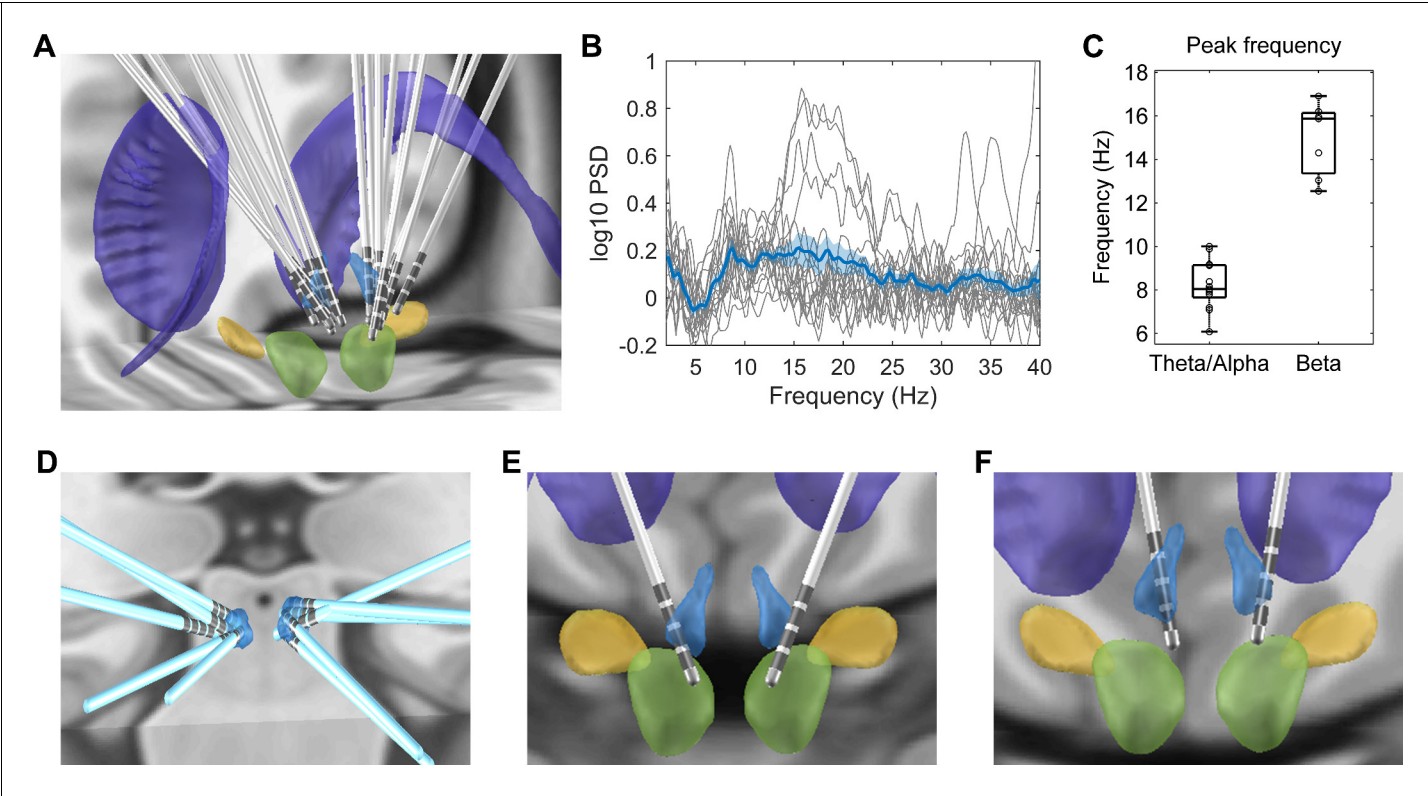

**Figure 2.** Electrode location and spectral characteristics of local field potentials from recorded habenula at rest. (A) Electrode locations reconstructed using Lead-DBS, with the structures colored in light blue for the habenula, purple for the caudate nucleus, light green for the red nucleus, and yellow for subthalamic nucleus. (B) The log-transformed oscillatory power spectra fitted using *fooof* method (after removing the non-oscillatory 1/f components). The bold blue line and shadowed region indicates the mean ± SEM across all recorded hemispheres, and the thin gray lines show measurements from individual hemispheres. (C) Boxplot showing the peak frequencies at theta/alpha and beta frequency bands from all recorded habenula. (D) Positions of the electrodes with theta peaks only during rest. (E) Electrode positions for Case 3, in whom only beta band peaks were detected in the resting activities from both sides. (F) Electrode positions for Case 6, in whom both theta and beta band peaks were present in resting activities from both sides.

The online version of this article includes the following source data for figure 2:

**Source data 1.** Source data for generating *Figure 2B, C*.

**Table 1.** Characteristics of enrolled subjects.

| Patient | Sex | Age (years) | Duration (years) | Disease | HAMD score | BDI score | Resting oscillation peaks | |
|---------|-----|-------------|------------------|---------|------------|-----------|---------------------------|---|
| | | | | | | | L | R |
| 1 | M | 21 | 5 | Schiz | NA | 32 | 9.1 Hz | 9.8 Hz |
| 2 | M | 21 | 5 | Dep | 12 | 10 | 7.9 Hz | 8.4 Hz |
| 3 | M | 44 | 10 | Bipolar | 23 | 22 | 14.3 Hz | 15.9 Hz |
| 4 | F | 19 | 4 | Schiz | NA | NA | 10 Hz | 8.1 Hz |
| 5 | M | 21 | 3 | Dep | 24 | 38 | 7.1 Hz | 16.9 Hz |
| 6 | M | 16 | 2 | Schiz | NA | 34 | 9.2 Hz; 13.0 Hz | 7.2 Hz; 12.5 Hz |
| 7 | F | 30 | 8 | Bipolar | 21 | 33 | 6.1 Hz | 7.8 Hz |
| 8 | F | 28 | 13 | Dep | 28 | 37 | No peak | 8.0 Hz |
| 9 | M | 35 | 20 | Dep | 25 | 34 | 16.2 Hz | 7.9 Hz; 16.0 Hz |

Hab: habenula; F: female; M: male; Dep: depressive disorder; Bipolar: bipolar disorder; Schiz: schizophrenia; HAMD: Hamilton Depression Rating Scale (17 items); BDI: Beck Depression Inventory; Both HAMD and BDI were acquired before the surgery. NA: not available.

### Transient theta/alpha activity in the habenula is differentially modulated by stimuli with positive and negative emotional valence

The power spectra normalized to the baseline activity (−2000 to −200 ms) showed a significant event-related synchronization (ERS) in the habenula spanning across 2–30 Hz from 50 to 800 ms after the presentation of all stimuli ($p_{cluster}$ < 0.05, *Figure 3A–C*). Permutation tests were applied to the power spectra in response to the negative and positive emotional pictures from all subjects. This identified two clusters with significant difference for the two emotional valence conditions: one in the theta/alpha range (5–10 Hz) at short latency (from 100 to 500 ms, *Figure 3D, E*) after stimulus presentation and another in the theta range (4–7 Hz) at a longer latency (from 2700 to 3300 ms, *Figure 3D, F*), with higher increase in the identified frequency bands with negative stimuli compared to positive stimuli in both clusters. The power of the activity at the identified frequency band for the neutral condition sits between the values for the negative condition and positive condition in both identified time windows (*Figure 3G, H*). It should be noted that there was an increase in a broadband activity at short latency (from 100 to 500 ms) after the stimuli onset (*Figure 3A–C*). This raises the question as to whether the emotional valence-related modulation observed in *Figure 3D*, especially the cluster at short latency, reflects a modulation of oscillations, which is not phase-locked to stimulus onset, or, alternatively, is it attributable to an evoked event-related potential (ERP). To address this question, we quantified the ERP for each emotional valence condition for each habenula. There was no significant difference in ERP latency or amplitude caused by different emotional valence stimuli (*Figure 3—figure supplement 1*). In addition, when only considering the non-phase-locked activity by removing the ERP from the time series before frequency-time decomposition, the emotional valence effect (presented in *Figure 3—figure supplement 2*) is very similar to those shown in *Figure 3*. These additional analyses demonstrated that the emotional valence effect in the LFP signal is more likely to be driven by non-phase-locked (induced only) activity, even though the possibility of the contribution from transient evoked potentials still cannot be completed excluded. Therefore, we refer to the activities in the habenula LFPs that are modulated by emotional valence at short latency after stimulus onset as 'activity' rather than 'oscillations'.

### Theta/alpha activity in the prefrontal cortex is also differentially modulated by stimuli with positive and negative emotional valence

For cortical activities measured using MEG, we first computed the time-frequency power spectra normalized to the baseline activity (−2000 to −200 ms) averaged across all MEG frontal sensors highlighted in *Figure 4A* for different stimulus emotional valence conditions for each recorded participant. The average power spectra across eight participants for different valence conditions are shown in *Figure 4B*. Permutation test applied to the power spectra in response to the negative and positive emotional pictures from all subjects identified clusters with significant differences ($p_{cluster}$ < 0.05) in the theta/alpha range at short latency (from 100 to 500 ms after stimulus onset) (*Figure 4C*).

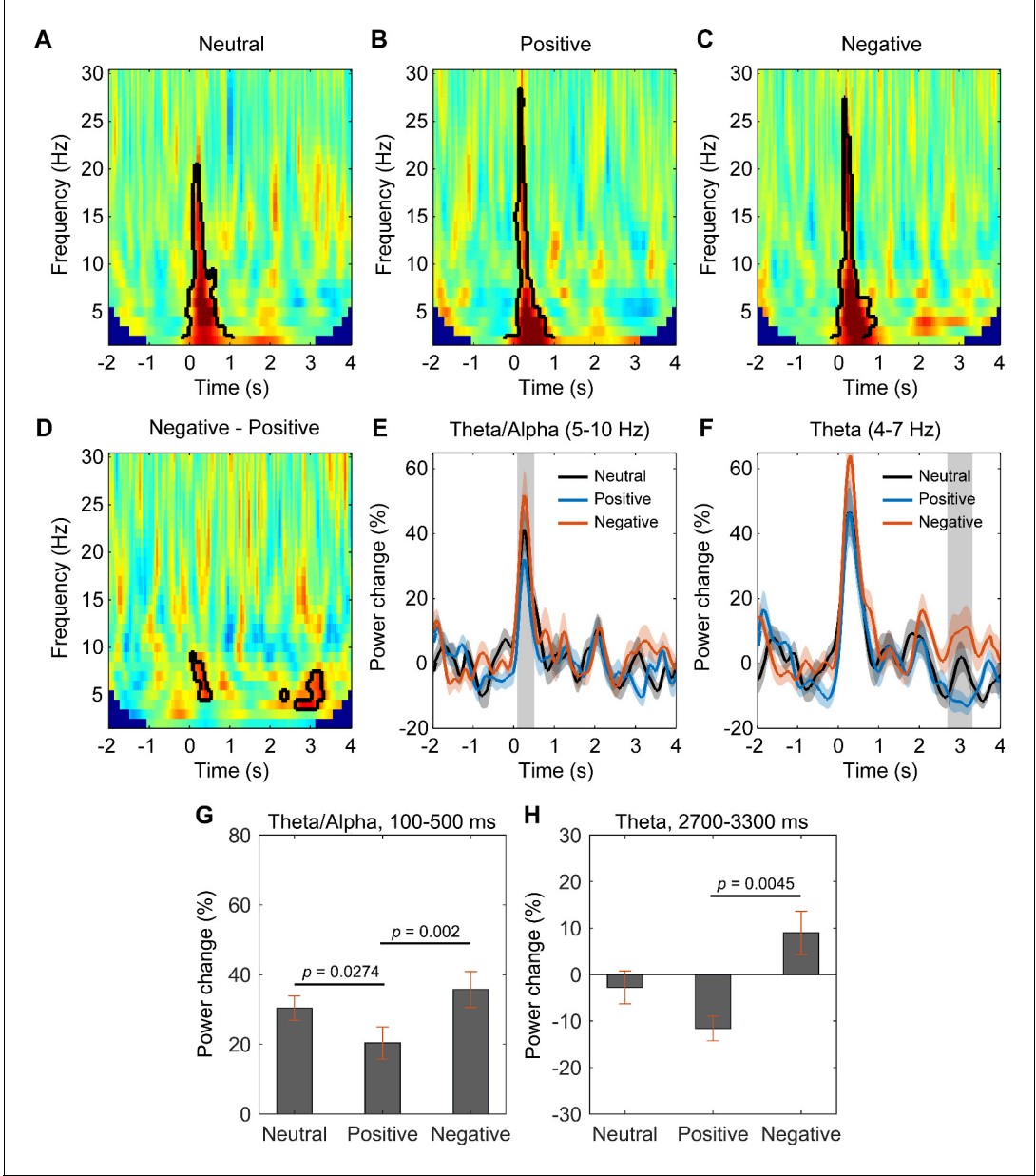

**Figure 3.** Habenular theta/alpha activity is differentially modulated by stimuli with positive and negative emotional valence (N = 18 habenula local field potential samples from nine subjects). (A–C) Time-frequency representations of the power response relative to pre-stimulus baseline (−2000 to −200 ms) for neutral (A), positive (B), and negative (C) valence stimuli, respectively. Significant clusters (p<0.05, non-parametric permutation test) are encircled with a solid black line. (D) Time-frequency representation of the power response difference between negative and positive valence stimuli, showing significant increased activity of the theta/alpha band (5–10 Hz) at short latency (100–500 ms) and another increased theta activity (4–7 Hz) at long latencies (2700–3300 ms) with negative stimuli (p<0.05, non-parametric permutation test). (E, F) Normalized power of the activities at theta/alpha (5–10 Hz) and theta (4–7 Hz) band over time. Significant difference between the negative and positive valence stimuli is marked by a shadowed bar (p<0.05, t-test corrected for multiple comparison). (G, H) The average spectral power relative to baseline activity in the identified time period and frequency band for different emotional valence conditions (5–10 Hz, 100–500 ms; 4–7 Hz, 2700–3300 ms). Significant difference was observed in theta/alpha power at 100–500 ms between neutral and positive condition (t-value = 2.4312, p=0.0274, 95% CI of the difference: [1.3203 18.6235]), between negative and positive condition (t-value = 4.5010, p=0.002, 95% CI of the difference: [8.0741 22.6561]), in theta power at 2700–3300 ms between negative and positive condition (t-value = 3.6944, p=0.0045, 95% CI of the difference: [8.8765 32.3104]).

The online version of this article includes the following source data and figure supplement(s) for figure 3:

**Source data 1.** Source data for generating *Figure 3*.

**Figure supplement 1.** Event-related potential (ERP) in habenula local field potential signals in different emotional valence (neutral, positive, and negative) conditions.

*Figure 3 continued on next page*

*Figure 3 continued*

**Figure supplement 2.** Non-phase-locked (induced only) activity in different emotional valence (neutral, positive, and negative) conditions (N = 18).

Subsequent cluster-based permutation statistical analysis of power changes over the identified frequency band (5–10 Hz) and time window (100–500 ms) confirmed significantly increased activity with negative stimuli in frontal sensors only (*Figure 4D*).

Next, we used a frequency-domain beamforming approach and statistics over eight subjects to identify the source of the difference in MEG theta/alpha reactivity within the 100–500 ms time window at the corrected significance threshold of p<0.05 with cluster-based permutation statistical analysis. We found two main significant source peaks with one in the right prefrontal cortex (PFC) (corresponding to Brodmann area 10, right superior frontal gyrus, MNI coordinate [16, 56, 0]; t-value = 4.14, p=0.046, corrected) (*Figure 5A*), and the other in the left PFC (corresponding to Brodmann area 9, left middle frontal gyrus, MNI coordinate [−32, 38, 28]; t-value = 3.21, p=0.046, corrected) (*Figure 5B*). No voxels within identified areas in *Figure 5* showed any significant

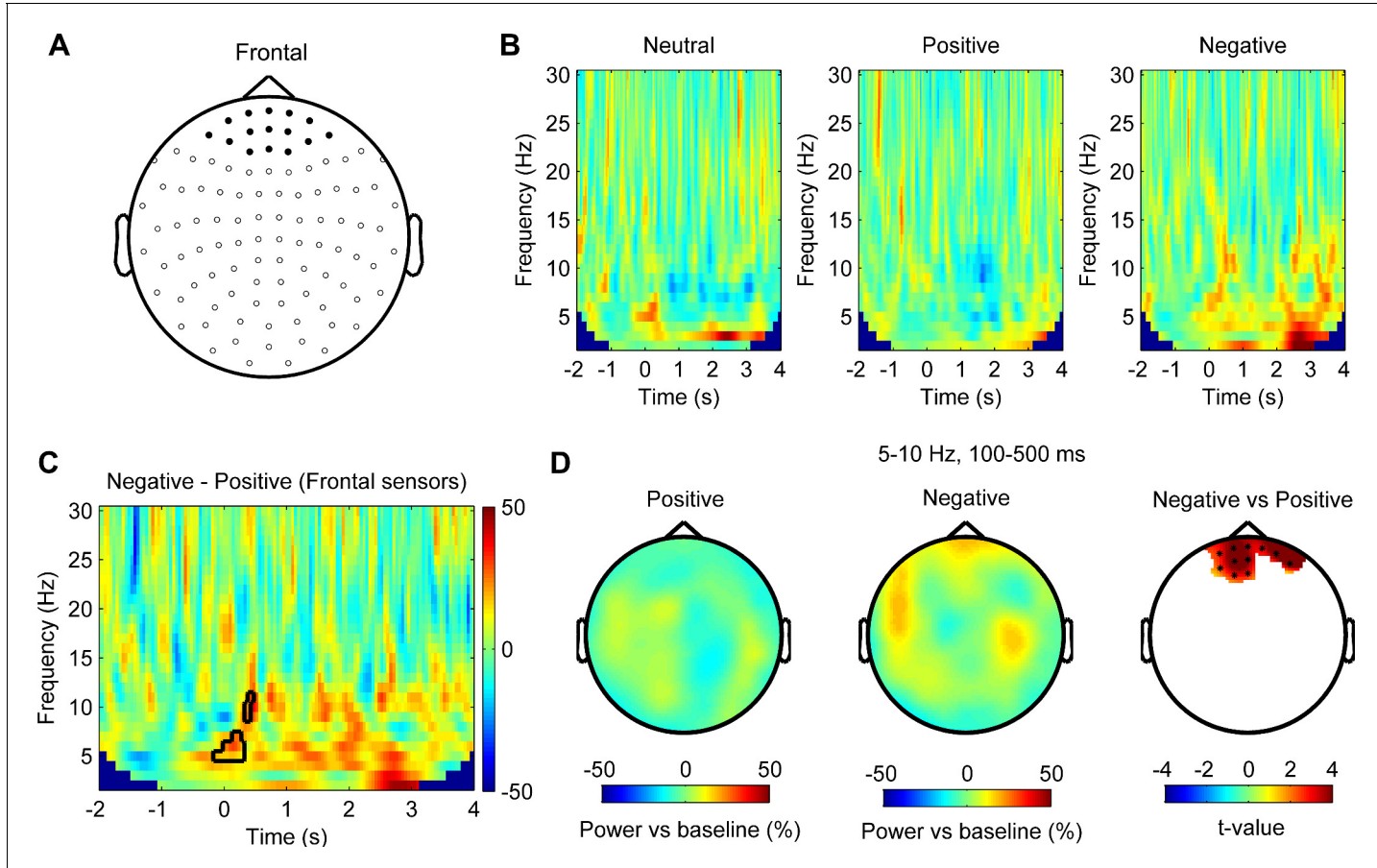

**Figure 4.** Theta/alpha oscillations in the prefrontal cortex are differentially modulated by stimuli with positive and negative emotional valence (N = 8 magnetoencephalography [MEG] samples from eight subjects). (**A**) Layout of the MEG sensor positions and selected frontal sensors (dark spot). (**B**) Time-frequency representation of the power changes relative to pre-stimulus baseline for neutral, positive, and negative stimuli averaged across frontal sensors (time 0 for stimuli onset). (**C**) Non-parametric permutation test showed clusters in the theta/alpha band at short latency after stimuli onset with significant difference (p<0.05) comparing negative and positive stimuli across frontal sensors. (**D**) Scalp plot showing the power in the 5–10 Hz theta/alpha band activity at 100–500 ms after the onset of positive (left), negative (middle) stimuli, and statistical t-values and sensors with significant difference (right) at a 0.05 significance level (corrected for whole-brain sensors).

The online version of this article includes the following source data for figure 4:

**Source data 1.** Source data for generating *Figure 4*.

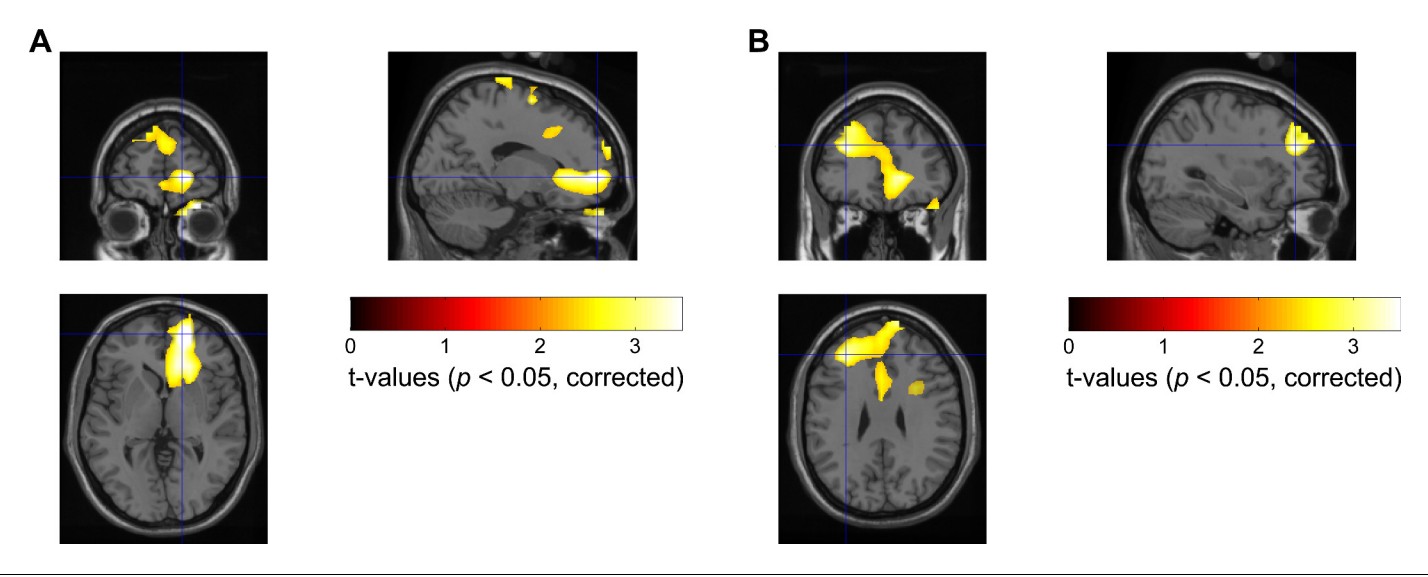

**Figure 5.** Statistical source maps of t-values (p<0.05; corrected for whole brain) for the comparison of magnetoencephalography (MEG) theta/alpha band (5–10 Hz) power reactivity to negative vs. positive emotional valence stimuli across subjects (N = 8 MEG samples from eight subjects). Dynamic imaging of coherent source beamformer was applied to the average theta/alpha band power changes from 100 to 500 ms after stimulus onset. The image was transformed to MNI template space and overlaid on the template structural image. The peak emotional valence-induced differences in the theta/alpha power were localized in the right Brodmann area 10, MNI coordinate [16, 56, 0] (shown in Plot A) and left Brodmann area 9, MNI coordinate [−32, 38, 28] (shown in Plot B).

The online version of this article includes the following source data and figure supplement(s) for figure 5:

**Source data 1.** Source data for generating *Figure 5*.

**Figure supplement 1.** An example of dynamic imaging of coherent source (DICS) beamforming for the movement-related beta power source localization in one participant (Case 8).

difference in the pre-cue baseline period, suggesting that the observed difference in the theta/alpha power reactivity was not due to difference in the baseline power between the two emotional valence conditions. Similar method has effectively identified the contralateral motor cortex as the source of modulation in the beta frequency band during button pressing movements, as shown in Figure 5—figure supplement 1.

## Cortical-habenular coherence is also differentially modulated by stimuli with positive and negative emotional valence

In addition, we asked how the coupling between habenula and cortex in the theta/alpha activity is modulated over time in the task and how the coupling changes with the valence of the presented stimuli. The cross-trial time-varying coherence between each MEG sensor and the habenula LFP was first calculated for each emotional valence condition, then averaged across all MEG sensors in each emotional valence condition for each habenula. Comparing the time-varying cortical-habenula coherence for the negative and positive emotional valence conditions across all recorded habenula (N = 16 from eight subjects with MEG recordings) showed increased coherence with negative stimulus in the theta/alpha band (5–10 Hz) in the time window of 800–1300 ms (paired t-test, df = 15, p<0.05, uncorrected, *Figure 6A*). We also performed the same analysis for cross-trial cortical-habenula coherence averaged across prefrontal channels and occipital channels separately. The emotional valence effect on the coherence was only observed in the frontal channels not in the occipital channels, as shown in *Figure 6—figure supplement 1*. Subsequent non-parametric cluster-based permutation statistical analysis of the coherence changes in this frequency band and selected time window (800–1300 ms) across the scalp revealed significantly increased coherence with negative stimuli over right frontal and temporal areas (N = 16, *Figure 6B*). Linear mixed-effect modeling confirmed significant effect of the increase in the theta/alpha band PFC-habenular coherence (relative to the pre-stimulus baseline) during this time window (800–1300 ms) on the theta power increase in the

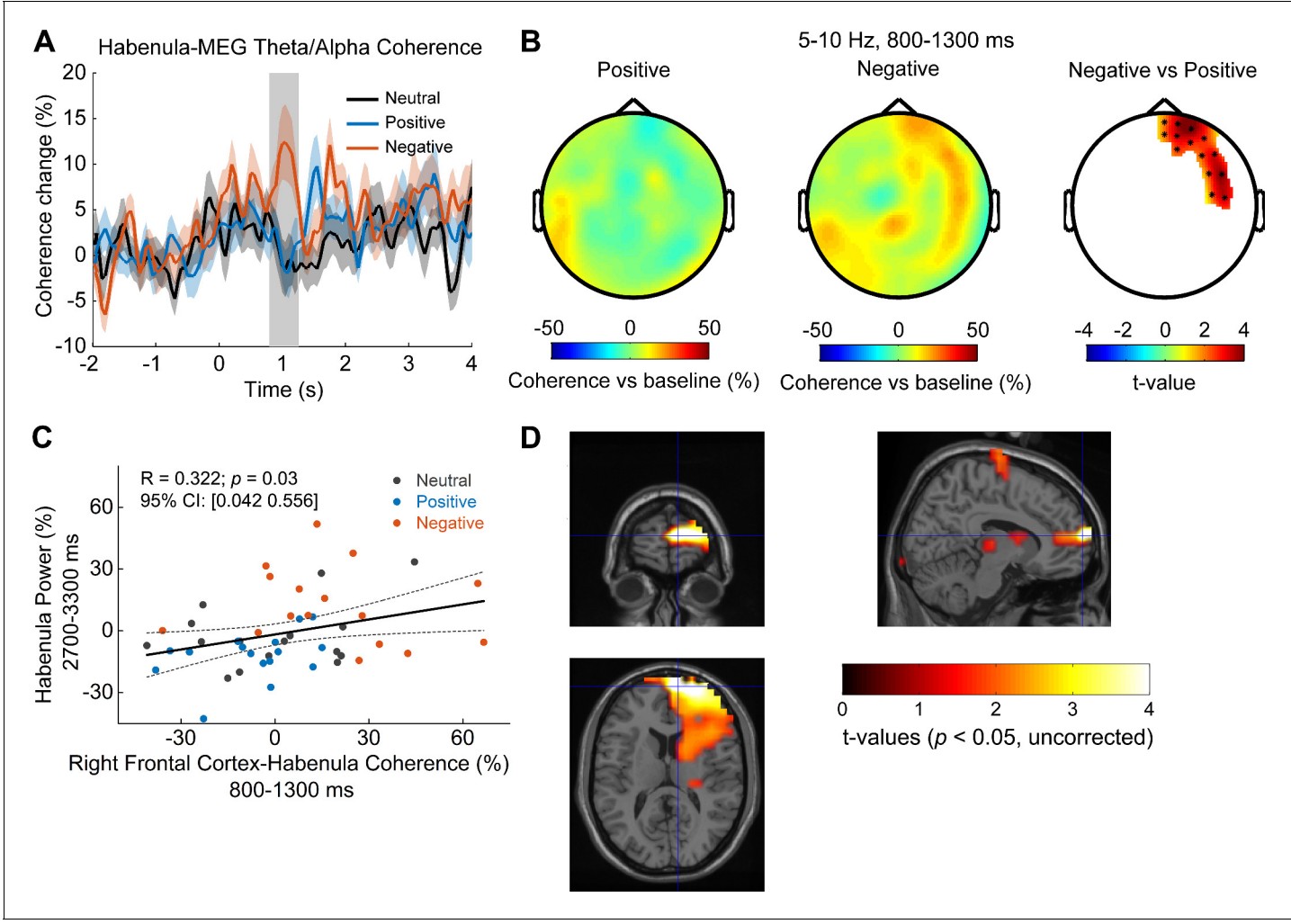

**Figure 6.** Cortical-habenular coherence in the theta/alpha band is also differentially modulated by stimuli with positive and negative emotional valence (N = 16 local field potential-magnetoencephalography [LFP-MEG] combination samples from eight subjects). (**A**) Time-varying theta (5–10 Hz) habenula-cortical coherence changes relative to pre-cue baseline averaged across all MEG channel combinations for each recorded habenula. The thick colored lines and shaded area show the mean and standard error across all recorded habenula. The coherence was significantly higher at 800–1300 ms after the onset of negative emotional stimuli compared to positive stimuli (rectangular shadow showing the time window with p<0.05). (**B**) Scalp plot showing the cortical-habenula coherence in the theta band during the identified time window (800–1300 ms) for positive stimuli (left), negative stimuli (middle), and statistical t-values and sensors with significant difference (right) masked at p<0.05 (corrected for whole-brain sensors). (**C**) The increase in the theta band coherence between right frontal cortex and habenula at 800–1300 ms correlated with the theta increase in habenula at 2700–3300 ms after stimuli onset. (**D**) Statistical source maps of t-values (p<0.05; uncorrected) for the comparison of theta/alpha coherence response in the time window of 800–1300 ms between negative stimuli with positive stimuli. The peak coherence differences were mainly localized in the right Brodmann area 10, MNI coordinate [10, 64, 12].

The online version of this article includes the following source data and figure supplement(s) for figure 6:

**Source data 1.** Source data for generating *Figure 6*.

**Figure supplement 1.** Time-varying coherence changes for frontal cortex-habenula and occipital cortex-habenula separately.

habenula at the later time window (2700–3300 ms after stimuli onset) (k = 0.2434 ± 0.1031 (95% confidence interval [0.0358 0.4509]), p=0.0226, $R^2$ = 0.104; *Figure 6C*). The non-parametric permutation test that is robust to outliers was also used to evaluate the correlation between the theta power and coherence when data from all participants and all emotional conditions were considered together. This confirmed significant correlation as well (R = 0.3224 (95% confidence interval: [0.0422 0.5557]), p=0.03). On the other hand, when data from different emotional conditions were considered separately, none of the separate correlations between theta coherence at 800–1300 ms and habenula theta power at 2700–3300 ms were significant: (R = 0.3405 (95% confidential interval: [−0.1867

0.7154]), p=0.2020 for neutral; R = 0.3846 (95% confidential interval: [−0.1373 0.7394]), p=0.1474 for positive; R = −0.1655 (95% confidential interval: [−0.6111 0.3597]), p=0.5474 for negative). In addition, we tested whether this coherence-power correlation was specific to the time window identified based on *Figure 6A*. To do so, we quantified the correlation between the habenula theta power at 2700–3300 ms and the habenula-PFC theta coherence at −200–300 ms, 300–800 ms, 1300–1800 ms, and 1800–2300 ms separately. None of the habenula-PFC coherences at other time windows correlated with habenula theta at 2700–3300 ms. We acknowledge that the effect shown in *Figure 6C* is weak and would not survive correction for multiple comparison. However, the selection of time window for the test shown in *Figure 6C* was based on the previous test shown in *Figure 6A*, not based on multiple tests.

Source localization of the theta/alpha habenula-cortical coherence difference for negative and positive stimuli revealed that theta/alpha coherence was higher with negative stimuli in right frontal regions, indicated in *Figure 6D*. The location of the peak t-statistic (t-value = 5.73, p=0.001, uncorrected) corresponds to MNI coordinate [10, 64, 12] and the region encompasses right medial PFC.

## Increased theta/alpha synchrony in the PFC-habenula network correlated with emotional valence, not arousal

It should be noted that there was co-variation between emotional valence and arousal in the stimuli presented (*Figure 1B*), and previous studies have shown that some neural activity changes in response to the viewing of affective pictures can be mediated by the effect of stimulus arousal (*Huebl et al., 2014*; *Huebl et al., 2016*). Therefore, we used linear mixed-effect modeling to assess whether the increased transient theta/alpha activity we observed in the habenula, the PFC, and in the PFC-habenula coherence in response to the viewing of negative compared to positive emotional pictures should be attributed to the emotional valence or the stimulus arousal. The nonlinear and non-monotonic relationship between arousal scores and the emotional valence scores shown in *Figure 1B* allowed us to differentiate the effect of the valence from arousal. The models identified significant fixed effects of valence on all the reported changes in the PFC-habenula network, but there was no effect of arousal (*Table 2* for the modeling and results). The negative effects of valence indicate that the lower the emotional valence score (more negative) of the presented stimuli, the higher the theta/alpha increase within the habenula, the PFC, and in the PFC-habenula theta band coherence, as shown in *Figure 7*.

Furthermore, we also investigated the relationship between the neural characteristics we observed and the clinical symptoms. However, none of the electrophysiological effects we observed correlated with clinical scores of depression (the Beck Depression Inventory score or Hamilton

**Table 2.** Linear mixed effect modeling details.

| ID | Model | Fixed effect of valence | | | Fixed effect of arousal | | | $R^2$ |
|---|---|---|---|---|---|---|---|---|
| | | k-Value | 95% CI | p-Value | k-Value | 95% CI | p-Value | |
| 1 | HabTheta1~ Valence+Arousal+ 1\|SubID | −2.8044 ± 0.9840 | [−4.7800,−0.8289] | 0.0063 | −2.5221 ± 2.5363 | [−7.6139, 2.5697] | 0.3247 | 0.6191 |
| 2 | HabTheta2~ Valence+Arousal+ 1\|SubID | −4.4526 ± 1.1753 | [−6.8121,−2.0932] | 0.0004 | 0.1975 ± 3.0295 | [−5.8844, 6.2794] | 0.9483 | 0.2557 |
| 3 | PFC_Theta~ Valence+Arousal+ 1\|SubID | −2.8921 ± 1.0221 | [−4.9507,−0.8334] | 0.0069 | −3.6237 ± 2.6252 | [−8.9112, 1.6637] | 0.1743 | 0.4368 |
| 4 | rPFC_Hab_Coh~ Valence+Arousal+ 1\|SubID | −6.1031 ± 1.6785 | [−9.4837,−2.7225] | 0.0007 | 3.5242 ± 4.3112 | [−5.1589, 12.2074] | 0.4180 | 0.2766 |

HabTheta1: theta/alpha band (5–10 Hz) in habenula LFPs at 100–500 ms ; HabTheta2: theta band (4–7 Hz) in habenula LFPs at 2700–3300 ms; PFC_Theta: theta/alpha band (5–10 Hz) averaged across frontal sensors at 100–500 ms; rPFC_Hab_Coh: theta/alpha band (5–10 Hz) coherence between right PFC and habenula at 800–1300 ms; Valence: valence value for the displayed pictures (1 = unpleasant -> 5 = neutral -> 9 = pleasant); Arousal: arousal value of the displayed pictures (1 = calm -> 9 = exciting); LFP: local field potential; PFC: prefrontal cortex.

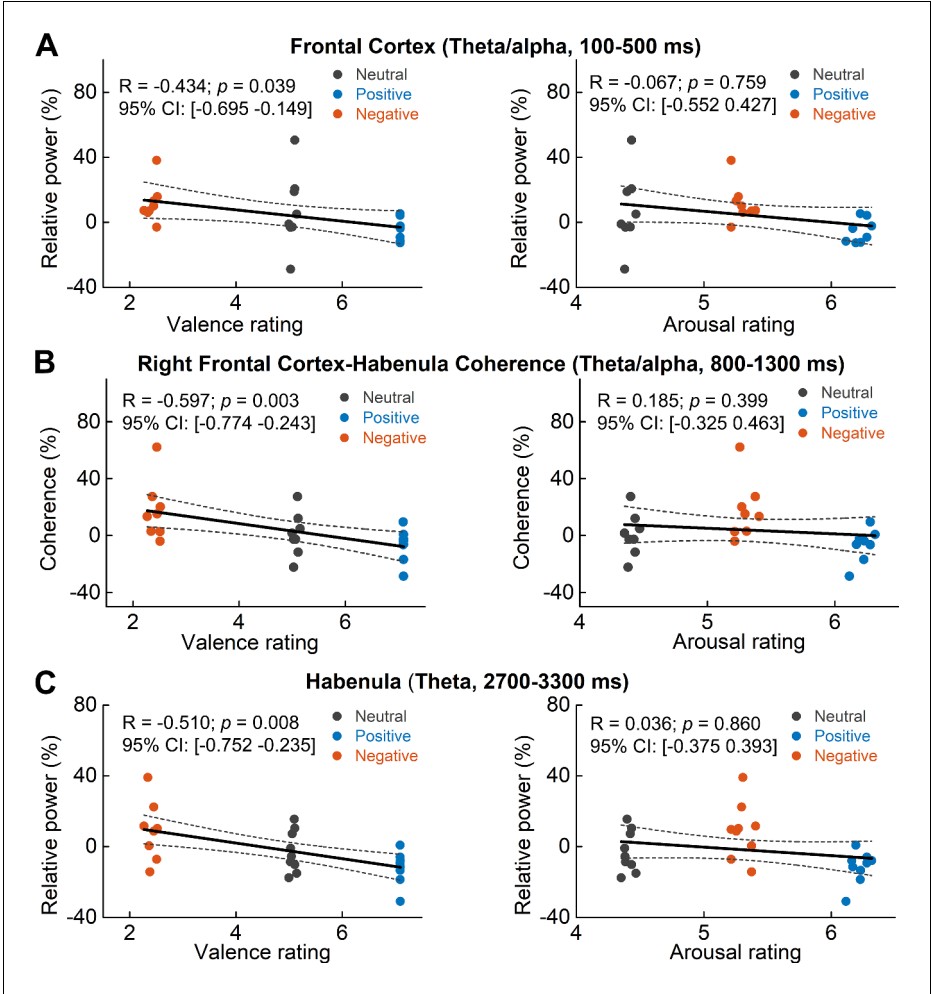

**Figure 7.** Scatter plots showing how early theta/alpha band power increase in the frontal cortex (**A**) theta/alpha band frontal cortex-habenula coherence (**B**) and theta band power increase at a later time window in habenula (**C**) changed with emotional valence (left column) and arousal (right column). Each dot shows the average of one participant in each categorical valence condition, which are also the source data of the multilevel modeling results presented in *Table 2*. The estimated correlation coefficient R and 95% confidence interval (CI), as well as the p value in the figure, are the results of partial correlation considering all data points together.
The online version of this article includes the following source data for figure 7:

**Source data 1.** Source data for generating *Figure 7* and *Table 2*.

Depression Rating Scale score) measured before the surgery across patients after correcting for multiple correction.

## Discussion

This study has showed that neural activities in the theta/alpha frequency band within the habenula and prefrontal cortical regions, as well as the connectivity between these structures in the same frequency band, are modulated in an emotional picture-viewing task in human participants. Compared with positive emotional stimuli, negative emotional stimuli were associated with higher transient increase in theta/alpha activity in both habenula and bilateral frontal cortex with a short latency (from 100 to 500 ms) after stimulus onset. Furthermore, higher theta/alpha coherence between habenula and right PFC was observed at 800–1300 ms after the stimulus onset, which was correlated with another increase in theta power in the habenula with a long latency (from 2700 to 3300 ms) after stimulus onset. These changes correlated with the emotional valence but not with the stimulus

arousal of the presented figures. These activity changes at different time windows may reflect the different neuropsychological processes underlying emotion perception including identification and appraisal of emotional material, production of affective states, and autonomic response regulation and recovery (*Phillips et al., 2003a*). The later effects of increased theta activities in the habenula when the stimuli disappeared were also supported by other literature showing that there can be prolonged effects of negative stimuli in the neural structure involved in emotional processing (*Haas et al., 2008*; *Puccetti et al., 2021*). In particular, greater sustained patterns of brain activity in the medial PFC when responding to blocks of negative facial expressions were associated with higher scores of neuroticism across participants (*Haas et al., 2008*). Slower amygdala recovery from negative images also predicts greater trait neuroticism, lower levels of likability of a set of social stimuli (neutral faces), and declined day-to-day psychological well-being (*Schuyler et al., 2014*; *Puccetti et al., 2021*). This is the first study, to our knowledge, implicating increased theta band activities in the habenula-PFC network in negative emotions in human patients.

## Habenula theta/alpha activity in negative emotional processing and major depression

The LHb has shown consistent hyperactivity in multiple animal models of depression-like phenotypes (*Hu et al., 2020*). Increased LHb activities have been observed during omission of a predicted reward, depressive-like phenotype, fear or stress (*Matsumoto and Hikosaka, 2009*; *Bromberg-Martin and Hikosaka, 2011*; *Wang et al., 2017*). Furthermore, manipulations enhancing or suppressing LHb activity in rodents lead to depressive-like or antidepressant effects, respectively (*Li et al., 2013*; *Lecca et al., 2016*; *Cui et al., 2018*; *Yang et al., 2018a*). Increased activation of the LHb inhibits dopamine neurons (*Ji and Shepard, 2007*; *Hikosaka, 2010*) and allows avoidance of threatening or unpleasant confrontations (*Shumake et al., 2010*; *Friedman et al., 2011*). In accordance with findings in animal models, several studies have provided evidence for habenula hyperactivity in human subjects with depressive disorders (*Morris et al., 1999*; *Lawson et al., 2017*).

To our knowledge, this is the first study showing increased neural activity in the habenula in the theta/alpha frequency band with perception of negative emotion in human participants. This is consistent with previous findings that LHb neurons in rodents in the depressive-like state showed increased firing with a mean firing rate in the theta frequency band (*Li et al., 2011*), and that ketamine reversed both the increase in theta activity in the habenula and depressive-like behavior in rodents (*Yang et al., 2018a*). The results in this study are also consistent with recent research showing that acute 5 Hz DBS of the LHb is associated with depressive-like behavior such as increased duration of immobility in a forced swim test in rodents (*Jakobs et al., 2019*). Possibly due to the limited sample size, we did not observe any correlation between the habenula theta/alpha activities and the Beck Depression Inventory score or Hamilton Depression Rating Scale score measured before the surgery across patients in this study. It therefore remains to be established whether hypersynchrony in the theta band in habenula might be associated with the development of depressive symptoms in human patients.

## Prefrontal cortex-habenular coherence in negative emotional processing

Apart from increased theta/alpha band synchronization within the bilateral habenula and PFC, our data showed that negative emotional stimuli induced increased theta/alpha coherence between the habenula and the right PFC. The increased rPFC-habenular coherence correlated with further increase of theta activities within the habenula at a later latency. These results suggest a specific role of the theta/alpha synchronization between habenula and frontal cortex in the perception of negative emotional valence. Previous studies have showed that LHb receives input from cortical areas processing information about pain, loss, adversities, bad, harmful, or suboptimal choices, such as the anterior insula and dorsal ACC (dACC) and the pregenual ACC (pgACC) (*Vadovičová, 2014*). Our data is consistent with the hypothesis that PFC-to-habenular projections provide a teaching signal for value-based choice behavior, helping to learn to avoid potentially harmful, low-valued or wrong choices (*Vadovičová, 2014*).

Our data also showed that the increase of PFC-habenular coherence during the presentation of negative emotional stimuli was mainly located in the right frontal cortex. Many studies have

investigated how both hemispheres have a role in emotional processing. The Right Hemisphere Hypothesis (RHH) suggests that the right hemisphere would be involved, more than the left hemisphere, in the processing of all emotional stimuli, irrespective of their emotional valence (*Gainotti, 2012*). On the other hand, the Valence Hypothesis (VH) posits that the left and the right hemispheres would be specialized in processing positive and negative emotions, respectively (*Davidson, 1992*; *Wyczesany et al., 2018*). The latter hypothesis has also been supported by studies of brain lesion (*Starkstein et al., 1987*; *Morris et al., 1999*), electroencephalography (EEG) (*Davidson, 1992*; *Wyczesany et al., 2018*), transcranial magnetic stimulation (TMS) (*Pascual-Leone et al., 1996*), and functional neuroimaging (*Canli et al., 1998*; *Beraha et al., 2012*) in the PFC. Our findings suggest a more important role of the functional connectivity between the right frontal cortex and habenula for the processing of negative emotions.

## Implications for the development of DBS therapy

Although the exact underlying physiological mechanism of DBS remains elusive, high-frequency DBS delivered to STN and GPi can reduce the firing rates of local neurons (*Boraud et al., 1996*; *Welter et al., 2004*) and suppress the hypersynchrony of oscillatory activities in the beta frequency band in the network, leading to symptom alleviation (*Kuhn et al., 2008*; *Oswal et al., 2016*) for Parkinson's disease. In addition, high-frequency DBS may also dissociate input and output signals, resulting in the disruption of abnormal information flow through the stimulation site (*Chiken and Nambu, 2016*). This is supported by recent studies showing that patient-specific connectivity profiles between the stimulation target and area of interest in the cortex can predict clinical outcome of DBS for Parkinson's disease (*Horn et al., 2017*), major depressive disorder (MDD) (*Riva-Posse et al., 2014*), and obsessive-compulsive disorder (*Baldermann et al., 2019*). Our results suggest that increased theta activity in the habenula and increased theta/alpha coherence between PFC and habenula are associated with negative emotional valence in human patients. High-frequency DBS targeting habenula may be beneficial for treatment-resistant MDD by inhibiting possible hyperactivity and theta band over-synchrony of neuronal activities in the habenula and by disrupting the information flow from the PFC to other midbrain areas through the habenula. It remains to be explored whether theta band synchronization can be used as a biomarker for closed loop habenula DBS for better treatment of MDD.

## Limitations

The response to emotional tasks is likely to be altered in patients with pathological mood states compared to healthy subjects. This study cannot address whether the emotional valence effect we observed is specific to psychiatric disorders or is a common feature of healthy emotional processing. Another caveat we would like to acknowledge is that the human habenula is a small region. Existing data from structural MRI scans reported combined habenula (the sum of the left and right hemispheres) volumes of ~ 30–36 mm$^3$ (*Savitz et al., 2011a*; *Savitz et al., 2011b*), meaning that each habenula has the size of 2–3 mm in each dimension, which may be even smaller than the standard functional MRI voxel size (*Lawson et al., 2013*). The size of the habenula is also small relative to the standard DBS electrodes (as shown in *Figure 2A*). The electrodes used in this study (Medtronic 3389) have electrode diameter of 1.27 mm with each contact length of 1.5 mm and contact spacing of 0.5 mm. We have tried different ways to confirm the location of the electrode and select the contacts that are within or closest to the habenula: (1) the MRI was co-registered with a CT image (General Electric, Waukesha, WI, USA) with the Leksell stereotactic frame to obtain the coordinate values of the tip of the electrode and (2) postoperative CT was co-registered to preoperative T1 MRI using a two-stage linear registration using Lead-DBS software. We used bipolar signals constructed from neighboring macroelectrode recordings, which have been shown to detect locally generated potentials from subthalamic nucleus and especially when the macroelectrodes are inside the subthalamic nucleus (*Marmor et al., 2017*). Considering that not all contacts for bipolar LFP construction are in the habenula in this study, as shown in *Figure 2*, we cannot exclude the possibility that the activities we measured are contaminated by activities from neighboring areas through volume conduction. In particular, the human habenula is surrounded by thalamus and adjacent to the posterior end of the medial dorsal thalamus, so we may have captured activities from the medial dorsal thalamus. However, we also showed that those bipolar LFPs from contacts in the habenula tend to have a peak in

the theta/alpha band in the power spectra density (PSD), whereas recordings from contacts outside the habenula tend to have extra peak in beta frequency band in the PSD. This supports the habenula origin of the emotional valence-related changes in the theta/alpha activities reported here. In addition, it should also be noted that a postoperative stun effect cannot be excluded, which could interfere with neural recordings, considering that the experiment took place only a few days after electrode implantation.

## Conclusion

In this study, we exploited the high temporal resolution of LFP and MEG measurements and observed an emotional valence effect in local activities and in cross-region coherence in the cortical-habenula network in different time windows. Our results provide evidence for the role of neural activity in the theta/alpha frequency band within the habenula and prefrontal cortical regions, as well as of theta/alpha coherence between these structures in the processing and experiencing of negative emotions in human patients.

# Materials and methods

## Participants

Nine patients (six males, aged 16–44, more details in *Table 1*) were recruited for this study, who underwent bilateral DBS surgery targeting the habenula as a clinical trial for treatment-resistant major depression (ClinicalTrials.gov identifier: NCT03347487) or as a pilot study for intractable schizophrenia or bipolar disorders. All participants gave written informed consent to the current study, which was approved by the local ethics committee of Ruijin Hospital, Shanghai Jiao Tong University School of Medicine, in accordance with the Declaration of Helsinki. The surgical procedure has been previously described (*Zhang et al., 2019*). The electrode position, stimulation parameters, and clinical outcome in Case 1 have been separately reported (*Wang et al., 2020*).

## DBS operation

Implantation of the quadripolar DBS electrodes (model 3389 [contact: 1.5 mm; distance: 0.5 mm; diameter: 1.27 mm]; Medtronic, Minneapolis, MN, USA) was performed under general anesthesia bilaterally using a MRI-guided targeting (3.0 T, General Electric). The MRI was co-registered with a CT image (General Electric, Waukesha) with the Leksell stereotactic frame to obtain the coordinate values (*Zhang et al., 2019*). The electrode leads were temporary externalized for 1 week.

## Paradigm

Patients were recorded in an emotional picture-viewing task (*Kuhn et al., 2005*; *Huebl et al., 2016*) 2–5 days after the first stage of the surgery for electrode implantation and prior to the second operation to connect the electrode to the subcutaneous pulse generator. During the task, participants were seated in the MEG scanner with a displaying monitor in front of them. Pictures selected from the Chinese Affective Pictures System (CAPS) (*Bai et al., 2005*) were presented on the monitor in front of them. The emotional valence (1 = unpleasant ⇒ 5 = neutral ⇒ 9 = pleasant) and arousal (1 = calm ⇒ 9 = exciting) of the pictures were previously rated by healthy Chinese participants (*Bai et al., 2005*). The figures can be classified into three valence categories (neutral, positive, and negative) according to the average score on emotional valence. As low-level properties of the figures, such as contrast brightness and complexity of the figures, are not measured or reported in the CAPS, only very dark or bright pictures were excluded from the paradigm. In our paradigm, each experiment consisted of multiple blocks of 30 trials, with each block including 10 pictures of each valence category (neutral, positive, and negative) in randomized order. Each trial started with a white cross ('+') presented with a black background for 1 s, indicating the participants to get ready and pay attention, then a picture was presented in the center of the screen for 2 s. This was followed by a blank black screen presented for 3–4 s (randomized). The task was programed using PsychoPy (https://www.psychopy.org/) with the timeline of each individual trial shown in *Figure 1A*. The participants were reminded to pay attention to the pictures displayed on the monitor and instructed to try to experience the emotions the pictures conveyed. An additional neutral picture was presented randomly three times per block, upon which the patients were supposed to press a button to ensure

constant attention during the paradigm. All participants completed 2–4 blocks of the paradigm and none of them missed any response to the additional figure, indicating that they kept focus and that their working memory required for the task is normal. Pictures displayed to different participants are overlapped but not exactly the same; the average valence and arousal values of the displayed pictures are as shown in *Figure 1B*. There were significant differences in the emotional valence scores, as well as in the arousal scores for the presented figures of the three emotional valence categories (one-way ANOVA followed by Bonferroni post hoc test, $F_{2,24}$ = 14642.02, p<0.0001 for the valence score, and $F_{2,24}$ = 2102.55, p<0.0001 for the arousal score). The positive figures have the highest valence scores and highest arousal scores; the negative figures have the lowest valence scores; whereas the neutral figures have the lowest arousal scores.

## Data acquisition

Whole-brain MEG and LFP were simultaneously recorded at a sampling frequency of 1000 Hz using a 306-channel, whole-head MEG system with integrated EEG channels (Elekta Oy, Helsinki, Finland). LFPs from all individual contacts (0, 1, 2, and 3, with 0 being the deepest contact) of the DBS electrodes were measured in monopolar mode with reference to a surface electrode attached to the earlobe or one of the most dorsal DBS contact. The MEG and LFP recordings were synchronized with the timing of the onset of each picture stimuli through an analogue signal sent by the laptop running the picture-viewing paradigm. The voltage of the analogue signal increased at the onset of the presentation of each picture and lasted for 500 ms before going back to zero. The voltage increase was different for pictures of different emotional valence category.

## Reconstruction of electrode locations in the habenula

We used the Lead-DBS toolbox (*Horn and Kühn, 2015*) to reconstruct the electrode trajectories and contact locations for all recorded patients (*Figure 2A*). Postoperative CT was co-registered to preoperative T1 MRI using a two-stage linear registration as implemented in Advanced Normalization Tools (ANT) (*Avants et al., 2008*). CT and MRI were spatially normalized into MNI_ICBM_2009b_N-LIN_ASYM space (*Fonov et al., 2011*). Electrodes were automatically pre-localized in native and template space using the PaCER algorithm (*Husch et al., 2018*), manually localized based on postoperative CT, and then warped into template (*Horn and Kühn, 2015*; *Horn et al., 2019*).

## LFP and MEG data analysis

All data were analyzed using MATLAB (R2013b) with FieldTrip (version 20170628) (*Oostenveld et al., 2011*) and SPM8 toolboxes. Bipolar LFP recordings were constructed offline by subtracting the monopolar recordings from neighboring contacts on each electrode. One bipolar LFP channel within or closest to the habenula was selected from each recorded hemisphere for final analysis based on the postoperative imaging data and the location reconstruction based on Lead-DBS (*Figure 2A*). Artifacts due to movement, flat, and jump artifacts in the LFPs and MEG recordings were visually inspected and manually marked during the preprocessing with FieldTrip. For MEG signals, there were strong magnetic artifacts in some channels, especially the sensors overlying the location of the percutaneous extension wires under the skin. Therefore, MEG data were denoised with the spatiotemporal signal space separation (tSSS) method (*Taulu and Simola, 2006*) with a 10 s time window and a subspace correlation limit of 0.9 using MaxFilter software (Elekta Neuromag Oy, version 2.2). All the selected bipolar LFPs and MEG recordings were high-pass filtered at 0.3 Hz, notch-filtered at 50 Hz and higher-order harmonics, low pass filtered at 100 Hz and then down-sampled to 250 Hz before further analysis. Eye blink and heartbeat artifacts in the MEG signals were identified by ICA, and the low-frequency, high-amplitude components were removed from all MEG sensors. The MEG data of one subject (Case 4) had to be discarded due to severe artifacts across all MEG channels. Hence, all reported results with MEG data are based on eight subjects.

The oscillatory activities in the habenula LFPs during rest were first investigated. The power spectra were calculated using the Fast Fourier Transform (FFT). We then applied the FOOOF algorithm (*Haller et al., 2018*; *Donoghue et al., 2020*) to separate the LFP power spectral densities into aperiodic (1/f-like component) and periodic oscillatory components, which are modeled as Gaussian peaks. With this algorithm, a periodic oscillatory component is detected only when its peak power exceeds that of aperiodic activity by a specified threshold. In this study, the algorithm was applied

to the 2–40 Hz range of the raw power spectra of the LFPs from each recorded hemisphere. We set the maximal number of power peaks (max_n_peaks) to be 4, the width of the oscillatory peak (peak_width_limits) to be between 1 and 15, and the threshold for detecting the peak (peak_threshold) to be 2. The goodness of fit was visually inspected for recordings from each hemisphere to make sure that the parameter settings worked well. After removing the aperiodic component, the periodic oscillatory components in the LFP power spectra were parameterized by their center frequency (defined as the mean of the Gaussian), amplitude (defined as the distance between the peak of the Gaussian and the aperiodic fit), and bandwidth (defined as two standard deviations of the fitted Gaussian) of the power peaks.

In the next step, we investigated the event-related power changes in the habenula LFPs and MEG signals in response to the presentation of figures of different emotional valence categories. All LFP and MEG signals were divided into event-related epochs aligned to the stimuli onset (−2500 to 4500 ms around the stimulus onset) and visually inspected for artifacts due to movement and other interferences. Trials with artifacts were removed from final analysis, leaving a mean number of 27 trials (range 18–30) for each valence category for each subject. A time-frequency decomposition using the wavelet transform-based approach with Morlet wavelet and cycle number of 6 was applied to each trial. We used a 500 ms buffer on both sides of the clipped data to reduce edge effects. The time-frequency representations were then averaged across trials of the same valence condition and baseline corrected to the average of pre-stimulus activity (−2000 to −200 ms) for each frequency band. Thus, resulting time-frequency values were percentage changes in power relative to the pre-stimulus baseline.

## MEG-specific data analysis

Statistical comparison of power over a determined frequency band and time window between stimulus conditions across the group of subjects was performed to find topographical space difference. MEG source localization was conducted using a frequency-domain beamforming approach. The dynamic imaging of coherent sources (DICS) beamformer in SPM8 with a single-shell forward model was used to generate maps of the source power difference between conditions on a 5 mm grid co-registered to MNI coordinates (*Gross et al., 2001*). A beamformer regularization parameter of 1% was used based on the previous study (*Litvak et al., 2010*). This beamforming method has been demonstrated to effectively suppress artifacts due to the presence of percutaneous extension wires and enable localization of cortical sources exhibiting stimuli-related power changes and cortical sources coherent with deep brain LFPs (*Litvak et al., 2010*; *Hirschmann et al., 2011*; *Litvak et al., 2011*). In this study, we focused our source analysis on the frequency band and time window identified by previous sensor-level power analysis to locate cortical sources of significant difference in the power response to negative and positive emotional stimuli.

## Cortical-habenular connectivity

The functional connectivity between habenula and cortical areas was investigated using coherence analysis, which provides a frequency-domain measure of the degree of co-variability between signals (*Litvak et al., 2010*; *Neumann et al., 2015*). The time-varying cross-trial coherence between each MEG sensor and the habenula LFP was first calculated for each emotional valence condition. For this, time-frequency auto- and cross-spectral densities in the theta/alpha frequency band (5–10 Hz) between the habenula LFP and each MEG channel at sensor level were calculated using the wavelet transform-based approach from −2000 to 4000 ms for each trial with 1 Hz steps using the Morlet wavelet and cycle number of 6. Cross-trial coherence spectra for each LFP-MEG channel combination were calculated for each emotional valence condition for each habenula using the function 'ft_connectivityanalysis' in Fieldtrip (version 20170628). Stimulus-related changes in coherence were assessed by expressing the time-resolved coherence spectra as a percentage change compared to the average value in the −2000 to −200 ms (pre-stimulus) time window for each frequency. Secondly, we determined the time window of interest by statistically comparing the sensor-level coherence between stimulus conditions. Third, cortical sources coherent with habenula-LFP activity in the determined frequency band and time window were located using DICS beamformer for each stimuli condition (*Gross et al., 2001*; *Litvak et al., 2011*).

## Statistics

A non-parametric cluster-based permutation approach (*Maris and Oostenveld, 2007*) was applied to normalized time-frequency matrices to identify clusters (time window and frequency band) with significant differences in the power changes induced by the presentation of pictures of different emotional valence. To achieve this, the original paired samples were randomly permuted 1000 times such that each pair was maintained but its assignment to the condition (negative or positive) may have changed to create a null-hypothesis distribution. For each permutation, the sum of the z-scores within suprathreshold clusters (pre-cluster threshold: $p < 0.05$) was computed to obtain a distribution of the 1000 largest suprathreshold cluster values. If the sum of the z-scores within a suprathreshold cluster of the original difference exceeded the 95th percentile of the permutation distribution, it was considered statistically significant. The average powers in the determined frequency band and time window identified by the cluster-based permutation method between different valence conditions were further compared using post hoc paired-sample permutation t-tests with the function in the PERMUTOOLS toolbox in MATLAB (*Crosse et al., 2020*). The t-value and associated confidence interval of the difference, as well as the p-value of the paired-sample permutation t-test, are reported.

A one-tailed dependent-sample t statistics and cluster-based permutation testing was applied to statistically quantify the differences in DICS source for power or source coherence between negative and positive emotional stimuli. In addition, linear mixed-effect modeling ('fitlme' in MATLAB) with different recorded subjects as random effects was used to investigate the correlations between the observed changes in the neural signals and to investigate whether any changes we observed in the neural activities were related to the ratings of the emotional valence or stimulus arousal of the stimuli. The estimated mean value, standard error, and 95% confidential interval of the fixed effect and associated p-values, as well as the $R^2$ value of the model were reported. The non-parametric permutation test that is robust to outliers was performed on Pearson's correlations using the PERMUTOOLS toolbox in MATLAB (*Crosse et al., 2020*) to further evaluate correlations when considering all data from different subjects together. The estimated R value with 95% confidence interval and estimated p-values of the test were reported.

## Acknowledgements

We would like thank Dr Wolf-Julian Neumann at Charité–University Medicine Berlin, Germany, for the discussion on the paradigm; Prof. Arjan Blokland at Maastricht University for the devices of behavioral task; Dr Ningfei Li at the Department for Neurology, Charité – University Medicine Berlin, Germany, for the discussion on Lead-DBS software and imaging; Dr Vladimir Litvak at Wellcome Centre for Human Neuroimaging, University College London, UK, for the discussion on MEG-LFP signal processing; and Yingying Zhang for the help of the emotion scaling.

## Additional information

### Funding

| Funder | Grant reference number | Author |
| --- | --- | --- |
| National Natural Science Foundation of China | 81571346 | Chunyan Cao |
| National Natural Science Foundation of China | 82071547 | Chunyan Cao |
| National Natural Science Foundation of China | 81771482 | Bomin Sun |
| Medical Research Council | MR/P012272/1 | Huiling Tan |
| Medical Research Council | MC_UU_12024/1 | Peter Brown Huiling Tan |
| National Institute for Health Research | | Peter Brown Huiling Tan |
| Rosetrees Trust | A1784 | Huiling Tan |

| University Challenge Seed Fund, Medical and Life Sciences Translational Fund, University of Oxford | UCSF 459 | Jean Debarros Huiling Tan |

The funders had no role in study design, data collection and interpretation, or the decision to submit the work for publication.

## Author contributions

Yongzhi Huang, Conceptualization, Data curation, Formal analysis, Methodology, Writing - original draft, Writing - review and editing; Bomin Sun, Resources, Funding acquisition, Investigation, Project administration, Writing - review and editing; Jean Debarros, Yijie Lai, Peter Brown, Formal analysis, Writing - review and editing; Chao Zhang, Shikun Zhan, Dianyou Li, Chencheng Zhang, Tao Wang, Peng Huang, Investigation, Writing - review and editing; Chunyan Cao, Conceptualization, Data curation, Formal analysis, Supervision, Funding acquisition, Investigation, Project administration, Writing - review and editing; Huiling Tan, Conceptualization, Data curation, Formal analysis, Supervision, Funding acquisition, Investigation, Writing - original draft, Project administration, Writing - review and editing

## Author ORCIDs

Yongzhi Huang (iD) https://orcid.org/0000-0002-2503-1589
Peter Brown (iD) http://orcid.org/0000-0002-5201-3044
Huiling Tan (iD) https://orcid.org/0000-0001-8038-3029

## Ethics

Clinical trial registration ClinicalTrials.gov Identifier: NCT03347487.
Human subjects: Informed consent, and consent to publish, was obtained before the recording. The study was approved by the local ethics committee of Ruijin hospital, Shanghai Jiao Tong University School of Medicine in accordance with the declaration of Helsinki.

## Decision letter and Author response

Decision letter https://doi.org/10.7554/eLife.65444.sa1
Author response https://doi.org/10.7554/eLife.65444.sa2

## Additional files

### Supplementary files

• Source code 1. The source code file is a compressed folder containing the MATLAB scripts to generate the figures and separate files for the source data to generate different figures with the file names indicating the figure or table with which the data was associated.

• Transparent reporting form

### Data availability

All data generated or analysed during this study are included in the manuscript and supporting files.

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
