## [Decision Letter]

**Acceptance summary:**

The findings provide novel insight into the network dynamics associated with the processing of emotional stimuli and in particular the role of the habenula. The key finding was a transient increase in habenula theta/alpha activity for negative compared to positive stimuli. Furthermore, there was a later increase in habenula-prefrontal oscillatory coupling in the same band. These are important data, as there are few reports of direct recordings from the habenula together with the MEG in humans performing cognitive tasks.

**Decision letter after peer review:**

Thank you for submitting your article "Increased theta/alpha synchrony in the habenula-prefrontal network with negative emotional stimuli in human patients" for consideration by *eLife*. Your article has been reviewed by 3 peer reviewers, including Ole Jensen as the Reviewing Editor and Reviewer #1, and the evaluation has been overseen by Floris de Lange as the Senior Editor. The following individuals involved in review of your submission have agreed to reveal their identity: Jan Hirschmann (Reviewer #2); Diego Vidaurre (Reviewer #3).

Essential Revisions:

Please pay special attention to

1) Concerns on volume conduction

2) The issue of evoked versus oscillatory responses

3) Concerns on low-level features of the visual stimuli driving the evoked effects

4) Robustness of findings in the light of saccadic artefacts and statistical assessment

*Reviewer #2 (Recommendations for the authors):*

Were the stimuli matched for low-level properties such as contrast and luminance? If not, I would suggest grouping them by e.g. luminance and comparing these groups with the same methods to test this potential confound.

The authors speak of a modulation of theta oscillations by emotional stimuli, yet the dominant structures in the time-frequency spectra of the habenula look more like an evoked response (wide band, low latency, inverted "V" shape). It might be worthwhile inspecting the stimulus-locked temporal average of the habenula signal and the time-frequency spectrum of that average to better understand the nature of this response.

In the discussion, it is mentioned that clinical scores and electrophysiological effects did not correlate. This important information should appear in the abstract and results, too.

It seems that the authors have used Medtronic extensions for externalization. Unless the custom-made non-magnetic version is used (please specify), these cause tremendous artifacts in the MEG signal. In this case, the authors should provide information on the degree of artifact contamination and the effectiveness of data cleaning.

In Figure 2A it seems that some electrodes were not placed in the habenula. Were these excluded? If not, why not?

Regarding my last comment in the public review about the somewhat unexpected timing of the coherence increase: Could this timing be an "artifact" of averaging over all MEG channels? Maybe this all-channel average mixes frontal and occipital coherence changes which themselves occur at more intuitive time points?

[Editors' note: further revisions were suggested prior to acceptance, as described below.]

Thank you for resubmitting your work entitled "Increased theta/alpha synchrony in the habenula-prefrontal network with negative emotional stimuli in human patients" for further consideration by *eLife*. Your revised article has been evaluated by Floris de Lange (Senior Editor) and a Reviewing Editor.

*Reviewer #1:*

The authors have adequately addressed most of my concerns. I do however remain concerned about whether the early transient activity in the theta/alpha band can be considered oscillatory given how short the effect is. I do appreciate that the authors now demonstrate that the activity is not only phase-locked to the stimulus, but that does not make it oscillatory per se. I suggest that the authors refer to this effect as transient theta/alpha band activity in the Abstract and though out the manuscript. I do also understand the coherence effect is later and more sustained and more like to reflect neuronal oscillations.

I realized it is not made clear how the correlations are calculated (Pearson I assume?). If so, are there concerns on outliers? Please clarify.

*Reviewer #2:*

My thanks to the authors for providing detailed and good answers to my comments. I have just two things left to say:

Regarding my comment 2: I am aware that the reported correlation includes all conditions, but based on the red dots in Figure 6C, it does not seem that the relationship is particularly strong for the "negative" condition. This can be checked by performing separate correlations for all three conditions. More importantly, I would ask the authors to repeat their power-coherence correlation for other time windows to check whether the correlation is specific to the time windows of interest. If it turns out the correlation is neither condition- nor time-specific, I would think that it reflects general similarity between power and coherence as well as temporal autocorrelation in both measures.

Regarding my comment 6: With the frontal channel selection we now see a difference between conditions in coherence emerging much earlier in the trial. To me, this shows that it is better use exactly the same channels/locations when presenting power and coherence.

*Reviewer #3:*

It's not my intention to be difficult, but it doesn't seem that the authors have addressed my concerns, or done any additional analyses to add statistical confidence on results that are based on a very low sample (inevitably, given the nature of the data).

To give an example, Figure 6 is exactly as before (except that now the number of subjects is noted in the caption). The authors have not followed my suggestion of performing nonparametric permutation testing on the correlation reported in Figure 6C, and it remains an open question whether this is driven by the blue dot at the bottom of the plot (potentially an outlier). Unfortunately, that other coherence measures (eg occipital – habenula) were not significant is not a valid argument; i.e. that does not give any information about whether this particular relationship is truly significant.

---

## [Author Response]

Essential Revisions:Please pay special attention to1) Concerns on volume conduction

Please see detailed response to Reviewer 1 Public Review Recommendation 1. In summary, we have tried different ways to confirm the location of the electrode and the contacts selected for analysis. We also tried to use bipolar recordings from the neighboring macroelectrodes in order to minimize activities from other areas due to volume conduction. However, considering the size of the habenula, the size of the macroelectrode contacts, and the resolution of the MRI, we still cannot completely exclude the possibility that our recordings can be contaminated by activities from other areas due to volume conduction. Therefore, we have now added in the following in the ‘Limitation’ of the Discussion:

“Another caveat we would like to acknowledge is that the human habenula is a small region. Existing data from structural MRI scans reported combined habenula (the sum of the left and right hemispheres) volumes of ~ 30–36 mm^3^ (Savitz et al., 2011a; Savitz et al., 2011b meaning that each habenula has the size of 2~3 mm in each dimension, which may be even smaller than the standard functional MRI voxel size (Lawson *et al.*, 2013). The size of the habenula is also small relative to the standard DBS electrodes (as shown in Figure 2A). The electrodes used in this study (Medtronic 3389) have electrode diameter of 1.27 mm with each contact length of 1.5 mm, and contact spacing of 0.5 mm. We have tried different ways to confirm the location of the electrode and to select the contacts that is within or closest to the habenula: 1.) the MRI was co-registered with a CT image (General Electric, Waukesha, WI, USA) with the Leksell stereotactic frame to obtain the coordinate values of the tip of the electrode; 2.) Post-operative CT was co-registered to pre-operative T1 MRI using a two-stage linear registration using Lead-DBS software. We used bipolar signals constructed from neighbouring macroelectrode recordings, which have been shown to detect locally generated potentials from subthalamic nucleus and especially when the macroelectrodes are inside the subthalamic nucleus (Marmor et al., 2017). Considering that not all contacts for bipolar LFP construction are in the habenula in this study, as shown in Figure 2, we cannot exclude the possibility that the activities we measured are contaminated by activities from neighbouring areas through volume conduction. In particular, the human habenula is surrounded by thalamus and adjacent to the posterior end of the medial dorsal thalamus, so we may have captured activities from the medial dorsal thalamus. However, we also showed that those bipolar LFPs from contacts in the habenula tend to have a peak in the theta/alpha band in the power spectra density (PSD); whereas recordings from contacts outside the habenula tend to have extra peak in beta frequency band in the PSD. This supports the habenula origin of the emotional valence related changes in the theta/alpha activities reported here.”

References:

Savitz JB, Bonne O, Nugent AC, Vythilingam M, Bogers W, Charney DS, et al. Habenula volume in post-traumatic stress disorder measured with high-resolution MRI. *Biology of Mood & Anxiety Disorders* 2011a; 1(1): 7.

Savitz JB, Nugent AC, Bogers W, Roiser JP, Bain EE, Neumeister A, et al. Habenula volume in bipolar disorder and major depressive disorder: a high-resolution magnetic resonance imaging study. *Biological Psychiatry* 2011b; 69(4): 336-43.

Lawson RP, Drevets WC, Roiser JP. Defining the habenula in human neuroimaging studies. *NeuroImage* 2013; 64: 722-7.

Marmor O, Valsky D, Joshua M, Bick AS, Arkadir D, Tamir I, et al. Local vs. volume conductance activity of field potentials in the human subthalamic nucleus. *Journal of Neurophysiology* 2017; 117(6): 2140-51.

2) The issue of evoked versus oscillatory responses

Please see detailed response to Reviewer 1 Public Review Recommendation 2. In summary, we have now quantified the evoked related potential (ERPs) for different emotional valence conditions separately for each subject, but we did not found emotional valence related modulation in the amplitude or the latency of the ERPs. In addition, we have also calculated induced oscillations only (after removing the evoked activity). And we found similar difference in the theta/alpha band activities caused by the emotional valence condition in the induced activities as we showed in the main results. So we argue that the difference in the early time window we reported in the manuscript is not dominated by evoked activity. The figures of the new analyses and following have now been added in the main text:

“In addition, we tested whether stimuli-related habenula LFP modulations primarily reflect a modulation of oscillations, which is not phase-locked to stimulus onset, or, alternatively, if they are attributed to evoked event-related potential (ERP). We quantified the ERP for each emotional valence condition for each habenula. There was no significant difference in ERP latency or amplitude caused by different emotional valence stimuli (Figure 3—figure supplement 1). In addition, when only considering the non phase-locked activity by removing the ERP from the time series before frequency-time decomposition, the emotional valence effect (presented in Figure 3—figure supplement 2) is very similar to those shown in Figure 3. These additional analyses demonstrated that the emotional valence effect in the LFP signal is more likely to be driven by non-phase-locked (induced only) activity.”

3) Concerns on low-level features of the visual stimuli driving the evoked effects

Please see detailed response to Reviewer 2 Recommendations for the authors 2. Unfortunately, low-level properties, such as contrast brightness and complexity of the figures are not measured or reported in the Chinese Affective Figure system. Only very dark or bright pictures were excluded from the paradigm. We have now added this information in the method (Line 367 – 369), and have also acknowledged this potential confounding in the discussion. However, we also expect that the effect of low-level properties in the figures, such as luminance or color, may have more prominent effect in the visual cortex measured from the parietal and occipital regions as shown in previous studies Yener et al., 2009; Haigh et al., 2018; Peter et al., 2019; Perry et al., 2020.

References:

Yener GG, Guntekin B, Tulay E, Basar E. A comparative analysis of sensory visual evoked oscillations with visual cognitive event related oscillations in Alzheimer's disease. *Neuroscience Letters* 2009; 462(3): 193-7.

Haigh SM, Cooper NR, Wilkins AJ. Chromaticity separation and the alpha response. *Neuropsychologia* 2018; 108: 1-5.

Peter A, Uran C, Klon-Lipok J, Roese R, van Stijn S, Barnes W, et al. Surface color and predictability determine contextual modulation of V1 firing and γ oscillations. *eLife* 2019; 8.

Perry G, Taylor NW, Bothwell PCH, Milbourn CC, Powell G, Singh KD. The γ response to colour hue in humans: Evidence from MEG. *PloS One* 2020; 15(12): e0243237.

4) Robustness of findings in the light of saccadic artefacts and statistical assessment

We have now addressed the concerns about the saccadic artefacts, please see the detailed response to the Reviewer 1 Public Review Recommendation 4.

We have now addressed the concerns about the robustness of the finding due to the average trial number per condition per participant, please see the response to the Reviewer 2 (Public Review Detailed Comment 1).

We have now addressed the concerns about the signal-to-noise ratio and ‘artefacts due to the extension cable for LFP recordings’, please see the response to Reviewer 2 (Recommendations for the authors, Point #4).

We have now addressed the concerns about the robustness of the statistics, please see the response to the Reviewer 3 Recommendation.

Reviewer #2 (Recommendations for the authors):Were the stimuli matched for low-level properties such as contrast and luminance? If not, I would suggest grouping them by e.g. luminance and comparing these groups with the same methods to test this potential confound.

We thank the reviewer for the comment. Unfortunately, low-level properties, such as contrast brightness and complexity of the figures are not measured or reported in the Chinese Affective Figure system. Only very dark or bright pictures were excluded from the paradigm. We have now added this information in the method: “As low-level properties of the figures, such as contrast brightness and complexity of the figures are not measured or reported in the CAPS, only very dark or bright pictures were excluded from the paradigm.” We have also acknowledged this potential confounding in the discussion. However, we also expect that the effect of low-level properties in the figures, such as luminance or color, may have more prominent effect in the visual cortex measured from the parietal and occipital regions as shown in previous studies Yener et al., 2009; Haigh et al., 2018; Peter et al., 2019; Perry et al., 2020.

References:

Yener GG, Guntekin B, Tulay E, Basar E. A comparative analysis of sensory visual evoked oscillations with visual cognitive event related oscillations in Alzheimer's disease. *Neuroscience Letters* 2009; 462(3): 193-7.

Haigh SM, Cooper NR, Wilkins AJ. Chromaticity separation and the alpha response. *Neuropsychologia* 2018; 108: 1-5.

Peter A, Uran C, Klon-Lipok J, Roese R, van Stijn S, Barnes W, et al. Surface color and predictability determine contextual modulation of V1 firing and γ oscillations. *eLife* 2019; 8.

Perry G, Taylor NW, Bothwell PCH, Milbourn CC, Powell G, Singh KD. The γ response to colour hue in humans: Evidence from MEG. *PloS One* 2020; 15(12): e0243237.

The authors speak of a modulation of theta oscillations by emotional stimuli, yet the dominant structures in the time-frequency spectra of the habenula look more like an evoked response (wide band, low latency, inverted "V" shape). It might be worthwhile inspecting the stimulus-locked temporal average of the habenula signal and the time-frequency spectrum of that average to better understand the nature of this response.

We agree with the reviewer that the activity increase in the first time window with short latency after the stimuli onset is very transient and not band-limited. This raise the question that whether this is oscillatory or a transient evoked activity. Please see the response to Reviewer 1 Recommendation 2 for the details of further analysis. In summary, we have now quantified the ERP but found no effect of emotional valence on the amplitude or latency of the ERP. We have also looked at the non-phase locked activities removing the influence of the ERP from the total power spectra, we didn’t observe any effect of the stimulus emotional valence in the pure induced (non-phase-locked) power spectra. Therefore, we argue that the theta/alpha increase with negative emotional stimuli we observed in both habenula and prefrontal cortex 0-500 ms after stimuli onset are not dominated by visual or other ERP.

In the discussion, it is mentioned that clinical scores and electrophysiological effects did not correlate. This important information should appear in the abstract and results, too.

Following the reviewer’s suggestion, we have now added the sentence in the results: “Furthermore, we also investigated the relationship between the neural characteristics we observed and the clinical symptoms. However, none of the electrophysiological effects we observed correlated with clinical scores of depression (the Beck Depression Inventory score or Hamilton Depression Rating Scale score) measured before the surgery across patients after correcting for multiple correction.” However due to the strict word limit in the abstract, we were not able to include this in the abstract.

It seems that the authors have used Medtronic extensions for externalization. Unless the custom-made non-magnetic version is used (please specify), these cause tremendous artifacts in the MEG signal. In this case, the authors should provide information on the degree of artifact contamination and the effectiveness of data cleaning.

Medtronic extensions for externalization was used in this study, NOT the custom-made non-magnetic version. The extension wires can reduce signal-to-noise ratio in the MEG recording. Therefore the spatiotemporal Signal Space Separation (tSSS) method (Taulu and Simola, 2006) implemented in the MaxFilter software (Elekta Oy, Helsinki, Finland) has been applied in this study to suppress strong magnetic artifacts caused by extension wires. This method has been proved to work well in de-noising the magnetic artifacts and movement artifacts in MEG data in our previous studies (Cao et al., 2019; Cao et al., 2020). In addition, the beamforming method proposed by several studies (Litvak et al., 2010; Hirschmann et al., 2011; Litvak et al., 2011) has been used in this study. In Litvak et al., 2010, the artifacts caused by DBS extension wires was described in detail and the beamforming was demonstrated to effectively suppress artifacts and thereby enable both localization of cortical sources coherent with the deep brain nucleus. We have now added more details and references about the data cleaning and the beamforming method in the main text. With the beamforming method, we did observe the standard movement-related modulation in the β frequency band in the motor cortex, shown in Figure 5 - figure supplement 1. This suggests that the beamforming method did work well to suppress the artefacts. The figure on movement-related modulation in the motor cortex in the MEG signals have now been added as a supplementary figure to demonstrate the effect of the beamforming.

References:

Taulu S, Simola J. Spatiotemporal signal space separation method for rejecting nearby interference in MEG measurements. *Physics in Medicine and Biology* 2006; 51(7): 1759-68.

Cao C, Huang P, Wang T, Zhan S, Liu W, Pan Y, et al. Cortico-subthalamic Coherence in a Patient With Dystonia Induced by Chorea-Acanthocytosis: A Case Report. *Frontiers in Human Neuroscience* 2019; 13: 163.

Cao C, Li D, Zhan S, Zhang C, Sun B, Litvak V. L-dopa treatment increases oscillatory power in the motor cortex of Parkinson's disease patients. *NeuroImage Clinical* 2020; 26: 102255.

Litvak V, Eusebio A, Jha A, Oostenveld R, Barnes GR, Penny WD, et al. Optimized beamforming for simultaneous MEG and intracranial local field potential recordings in deep brain stimulation patients. *NeuroImage* 2010; 50(4): 1578-88.

Litvak V, Jha A, Eusebio A, Oostenveld R, Foltynie T, Limousin P, et al. Resting oscillatory cortico-subthalamic connectivity in patients with Parkinson's disease. *Brain* 2011; 134(Pt 2): 359-74.

Hirschmann J, Ozkurt TE, Butz M, Homburger M, Elben S, Hartmann CJ, et al. Distinct oscillatory STN-cortical loops revealed by simultaneous MEG and local field potential recordings in patients with Parkinson's disease. *NeuroImage* 2011; 55(3): 1159-68.

In Figure 2A it seems that some electrodes were not placed in the habenula. Were these excluded? If not, why not?

We didn’t exclude any electrode shown in Figure 2A because we showed that those electrodes from which only alpha/theta peaks were detected are well placed in the habenula area based on the Lead-DBS software. However, we noticed that those electrodes from which large beta band peak were present in the power spectra density and seem to be outside the habenula, we can still detect another peak of smaller amplitude in the theta/alpha band. The human habenula is a small region. Existing data from structural MRI scans reported combined habenula (the sum of the left and right hemisphere) volumes of ~ 30–36 mm^3^ (Savitz et al., 2011a; Savitz et al., 2011b) which means each habenula has the size of 2~3 mm in each dimension, which may be even smaller than the standard functional MRI voxel size (Lawson et al. 2013). The Lead-DBS software gives a good indication on the location of the electrode relative to a standardised atlas of habenula: THOMAS Atlas (Su et al., 2019). But we feel that it might not be valid to use Lead-DBS results to provide the binary selection criteria for such a small structure, as the electrode outside the habenula as shown in Figure 2 also have contacts which are at most 1 or 2 pixels away from the habenula. We have now also acknowledged that we cannot exclude the possibility that the activities we measured are contaminated by activities from neighbouring areas through volume conduction. In particular, the human habenula is surrounded by thalamus and adjacent to the posterior end of the medial dorsal thalamus, so the β band activities captured by some electrodes may come from the medial dorsal thalamus. We have now added this as part of the ‘Limitation’. Please see detailed response to Reviewer 1 Comment 1.

References:

Savitz JB, Bonne O, Nugent AC, Vythilingam M, Bogers W, Charney DS, et al. Habenula volume in post-traumatic stress disorder measured with high-resolution MRI. *Biology of Mood & Anxiety Disorders* 2011a; 1(1): 7.

Savitz JB, Nugent AC, Bogers W, Roiser JP, Bain EE, Neumeister A, et al. Habenula volume in bipolar disorder and major depressive disorder: a high-resolution magnetic resonance imaging study. *Biological Psychiatry* 2011b; 69(4): 336-43.

Lawson RP, Drevets WC, Roiser JP. Defining the habenula in human neuroimaging studies. *NeuroImage* 2013; 64: 722-7.

Su JH, Thomas FT, Kasoff WS, Tourdias T, Choi EY, Rutt BK, et al. Thalamus Optimized Multi Atlas Segmentation (THOMAS): fast, fully automated segmentation of thalamic nuclei from structural MRI. *NeuroImage* 2019; 194: 272-82.

Regarding my last comment in the public review about the somewhat unexpected timing of the coherence increase: Could this timing be an "artifact" of averaging over all MEG channels? Maybe this all-channel average mixes frontal and occipital coherence changes which themselves occur at more intuitive time points?

We have now looked at the cross-trial coherence changes for frontal cortex-habenula and occipital cortex-habenula separately. When only MEG channels from the frontal cortex was considered, the coherence with habenula showed significant increase around 1 s which is very similar (Figure S7 related to Figure 6A) as we reported in the main manuscript. But the coherence first peaked at around 500 ms after stimuli onset, which is still after the initial activities increase we observed in both the prefrontal cortex and the habenula. In contrast, we didn’t observe any changes in the coherence between occipital channels and habenula. So the timing with significant difference in the prefrontal cortex-habenula area coherence is not an ‘artifact’ of averaging all MEG channels. The following has now been added in the main text: “We also performed the same analysis for cross-trial cortical-habenula coherence averaged across prefrontal channels and occipital channels separately. The emotional valence effect on the coherence was only observed in the frontal channels not in the occipital channels, as shown in Figure 6—figure supplement 1.”

[Editors' note: further revisions were suggested prior to acceptance, as described below.]

Reviewer #1:The authors have adequately addressed most of my concerns. I do however remain concerned about whether the early transient activity in the theta/alpha band can be considered oscillatory given how short the effect is. I do appreciate that the authors now demonstrate that the activity is not only phase-locked to the stimulus, but that does not make it oscillatory per se. I suggest that the authors refer to this effect as transient theta/alpha band activity in the Abstract and though out the manuscript. I do also understand the coherence effect is later and more sustained and more like to reflect neuronal oscillations.

We thank the reviewer for the comment. We have now made the revisions accordingly. The term “oscillatory activity” induced by stimulus has been changed to “transient activity” consistently through the abstract and main text, especially when referring to the modulation from 100 to 500 ms after the stimuli onset. In addition, we have now added the following text in the Results: “It should be noted that there was an increase in a broad band activity at short latency (from 100 to 500 ms) after the stimuli onset (Figure 3A-C). This raises the question as to whether the emotional valence related modulation observed in Figure 3D, especially the cluster at short latency, reflects a modulation of oscillations, which is not phase-locked to stimulus onset, or, alternatively, are they attributable to an evoked event-related potential (ERP).’. The following sentence has also been included: ‘These additional analyses demonstrated that the emotional valence effect in the LFP signal is more likely to be driven by non-phase-locked (induced only) activity, even though the possibility of the contribution from transient evoked potentials still cannot be completed excluded. Therefore, we refer to the activities in the habenula LFPs that are modulated by emotional valence at short latency after stimulus onset as ‘activity’ rather than ‘oscillations’.”

I realized it is not made clear how the correlations are calculated (Pearson I assume?). If so, are there concerns on outliers? Please clarify.

In the Figure 6C, the correlation was estimated by linear mixed-effect modelling method. Regarding the concerns on outliers, the nonparametric permutation test that is robust to outliers performed on the Pearson’s correlation using the PERMUTOOLS toolbox in MATLAB (https://github.com/mickcrosse/PERMUTOOLS) has now been used to evaluate correlation between the theta power and coherence. The new test has showed significant correlation between PFC-habenula theta coherence and habenula theta power as well (R = 0.3224 (95% confidence interval: [0.0422 0.5557]), *p* = 0.03). In addition, to address the comments from other reviewers, we have now applied the permutation based correlation test on data from different emotional valence conditions separately, and also applied the correlation test to the PFC-habenula coherence at other time windows. The following has now been added in the main text:

Methods: “The nonparametric permutation test that is robust to outliers was performed on the Pearson’s correlations using the PERMUTOOLS toolbox in MATLAB (https://github.com/mickcrosse/PERMUTOOLS) to further evaluate correlations when considering all data from different subjects together. The estimated R value with 95% confidence interval, and estimated *p* values of the test were reported.”

Results: “The nonparametric permutation test that is robust to outliers (https://github.com/mickcrosse/PERMUTOOLS) was also used to evaluate the correlation between the theta power and coherence when data from all participants and all emotional conditions were considered together. This confirmed significant correlation as well (R = 0.3224 (95% confidence interval: [0.0422 0.5557]), *p* = 0.03). On the other hand, when data from different emotional conditions were considered separately, none of the separate correlations between theta coherence at 800–1300 ms and habenula theta power at 2700–3300 ms were significant: (R = 0.3405 (95% confidential interval: [-0.1867 0.7154]), *p* = 0.2020 for Neutral; R = 0.3846 (95% confidential interval: [-0.1373 0.7394]), *p* = 0.1474 for Positive; R = -0.1655 (95% confidential interval: [-0.6111 0.3597]), *p* = 0.5474 for Negative). In addition, we tested whether this coherence-power correlation was specific to the time window identified based on Figure 6A. To do so, we quantified the correlation between the habenula theta power at 2700–3300 ms and the habenula-PFC theta coherence at -200–300 ms, 300–800 ms, 1300–1800 ms, and 1800–2300ms separately. None of the habenula-PFC coherences at other time windows correlated with habenula theta at 2700–3300 ms. We acknowledge that the effect shown in Figure 6C is weak and would not survive correction for multiple comparison. However, the selection of time window for the test shown in Figure 6C was based on the previous test shown in Figure 6A, not based on multiple tests.”

Reviewer #2:My thanks to the authors for providing detailed and good answers to my comments. I have just two things left to say:Regarding my comment 2: I am aware that the reported correlation includes all conditions, but based on the red dots in Figure 6C, it does not seem that the relationship is particularly strong for the "negative" condition. This can be checked by performing separate correlations for all three conditions. More importantly, I would ask the authors to repeat their power-coherence correlation for other time windows to check whether the correlation is specific to the time windows of interest. If it turns out the correlation is neither condition- nor time-specific, I would think that it reflects general similarity between power and coherence as well as temporal autocorrelation in both measures.

We thank the reviewer for the comments. We have now applied the permutation based correlation test on data from different emotional valence conditions separately, and also applied the correlation test to the habenula theta power and PFC-habenula coherence at other time windows. When the PFC-habenula theta band coherence at 800–1300 ms was considered, none of the separate correlations between PFC-habenula coherence and habenula theta power at 2700–3300 ms for different emotional valence conditions were significant (R^2^ = 0.1156, *p* = 0.1969 for neutral valence condition; R^2^ = 0.1475, *p* = 0.1413 for positive valence condition; R^2^ = 0.0274, *p* = 0.5402 for negative valence condition). Results were similar when nonparametric tests (https://github.com/mickcrosse/PERMUTOOLS) based on permutation were used: the test confirmed significant correlation when data from all conditions were considered together (R = 0.3224 (95% confidence interval: [0.0422 0.5557]), p = 0.03), but this was not the case for correlation when data from different emotional valence conditions were considered separately: R = 0.3405 (95% confidential interval: [-0.1867 0.7154]), *p* = 0.2020 for neutral; R = 0.3846 (95% confidential interval: [-0.1373 0.7394]), *p* = 0.1474 for positive; R = -0.1655 (95% confidential interval: [-0.6111 0.3597]), *p* = 0.5474 for negative condition.

We also investigated whether the PFC-habenula coherence at other time windows, such as -200–300 ms, 300–800 ms, 1300–1800 ms, and 1800–2300ms is correlated with habenula theta power at the later time window (around 3000 ms). The Linear mixed-effect modelling analysis showed that the power-coherence correlations for other time windows are not significant (Figure S1), indicating the correlation between power and coherence is time-specific.

We have now added these results to the main text:

Methods: “The nonparametric permutation test that is robust to outliers was performed on the Pearson’s correlations using the PERMUTOOLS toolbox in MATLAB (https://github.com/mickcrosse/PERMUTOOLS) to further evaluate correlations when considering all data from different subjects together. The estimated R value with 95% confidence interval, and estimated p values of the test were reported.”

Results: “The nonparametric permutation test that is robust to outliers (https://github.com/mickcrosse/PERMUTOOLS) was also used to evaluate the correlation between the theta power and coherence when data from all participants and all emotional conditions were considered together. This confirmed significant correlation as well (R = 0.3224 (95% confidence interval: [0.0422 0.5557]), *p* = 0.03). On the other hand, when data from different emotional conditions were considered separately, none of the separate correlations between theta coherence at 800–1300 ms and habenula theta power at 2700–3300 ms were significant: (R = 0.3405 (95% confidential interval: [-0.1867 0.7154]), *p* = 0.2020 for Neutral; R = 0.3846 (95% confidential interval: [-0.1373 0.7394]), *p* = 0.1474 for Positive; R = -0.1655 (95% confidential interval: [-0.6111 0.3597]), *p* = 0.5474 for Negative). In addition, we tested whether this coherence-power correlation was specific to the time window identified based on Figure 6A. To do so, we quantified the correlation between the habenula theta power at 2700–3300 ms and the habenula-PFC theta coherence at -200–300 ms, 300–800 ms, 1300–1800 ms, and 1800–2300ms separately. None of the habenula-PFC coherences at other time windows correlated with habenula theta at 2700–3300 ms. We acknowledge that the effect shown in Figure 6C is weak and would not survive correction for multiple comparison. However, the selection of time window for the test shown in Figure 6C was based on the previous test shown in Figure 6A, not based on multiple tests.”

Regarding my comment 6: With the frontal channel selection we now see a difference between conditions in coherence emerging much earlier in the trial. To me, this shows that it is better use exactly the same channels/locations when presenting power and coherence.

The earlier time window (from 500 to 600 ms) with significant difference between frontal MEG-habenula LFP coherence for conditions was very brief. In addition, this time window was not significant when the other coupling method (imaginary coherence) was used to evaluate the connectivity between frontal MEG and habenula (Figure S2). The time window in which the PFC-habenula connectivity was consistently modulated by emotional valence was only around 1000 ms (8000-1200 ms), but not at 500-600 ms. Therefore, we still focused on this time window for the other analysis shown in Figure 6.

Reviewer #3:It's not my intention to be difficult, but it doesn't seem that the authors have addressed my concerns, or done any additional analyses to add statistical confidence on results that are based on a very low sample (inevitably, given the nature of the data).

We are sorry for not addressing the question on statistical confidence in the previous version. We appreciate the concern of the reviewer given the small sample size we have. We have now added 95% confidence intervals into the main text in the ‘Results’. For example, we have now used a non-parametric bootstrap test for the post-hoc analysis shown in Figure 3 G and H, and have now shown the confidence interval of the test. We have now made the following changes in the main text: “The average powers in the determined frequency band and time window identified by the cluster-based permutation method between different valence conditions were further compared using post-hoc paired-sample permutation t-tests with the function in the PERMUTOOLS toolbox in MATLAB (https://github.com/mickcrosse/PERMUTOOLS). The t value and associated confidence interval of the difference, as well as the p value of the paired-sample permutation t-test are reported.”. The statistical results were updated in Figure 3.

In addition, we have now included the confidence interval in the effects of the multi-level modelling. For example, Page 9 Line 176: ‘k = 0.2434 ± 0.1031 (95% confidence interval [0.0358 0.4509])’ and the Table 2.

We have also added 95% confidential intervals for the results shown in Figure 7. The 95% CI for the partial correlation coefficient between valence rating and frontal theta/alpha power, between valence rating and coherence, between valence rating and habenula theta power, controlling for arousal rating are [-0.6946, -0.1491], [-0.7738, -0.2429], [-0.7519, -0.2353], respectively. The 95% confidence intervals for the partial correlation coefficient between arousal rating and frontal theta/alpha power, between arousal rating and coherence, between arousal rating and habenula theta power, controlling for valence rating are [-0.5517, 0.4273], [-0.3253, 0.4634], [-0.3748, 0.3926], respectively.

To give an example, Figure 6 is exactly as before (except that now the number of subjects is noted in the caption). The authors have not followed my suggestion of performing nonparametric permutation testing on the correlation reported in Figure 6C, and it remains an open question whether this is driven by the blue dot at the bottom of the plot (potentially an outlier). Unfortunately, that other coherence measures (eg occipital – habenula) were not significant is not a valid argument; i.e. that does not give any information about whether this particular relationship is truly significant.

Following the reviewer’s suggestion, the nonparametric permutation test that is robust to outliers has now been used based on the Pearson’s correlation between the theta power and coherence using the PERMUTOOLS toolbox in MATLAB (https://github.com/mickcrosse/PERMUTOOLS), and confirms significant coherence (R = 0.3224 (95% confidence interval: [0.0422 0.5557]), *p* = 0.03). In addition, the linear mixed-effect modelling analysis after removing the dot at the bottom of the plot showed significant correlation between the coherence and the theta power (R^2^ = 0.0832, *p* = 0.0447).

We have now added the following to the main text:

Methods: ‘The nonparametric permutation test that is robust to outliers was performed on the Pearson’s correlations using the PERMUTOOLS toolbox in MATLAB (https://github.com/mickcrosse/PERMUTOOLS) to further evaluate correlations when considering all data from different subjects together. The estimated R value with 95% confidence interval, and estimated p values of the test were reported.’

Results: ‘The nonparametric permutation test that is robust to outliers (https://github.com/mickcrosse/PERMUTOOLS) was also used to evaluate the correlation between the theta power and coherence when data from all participants and all emotional conditions were considered together. This confirmed significant correlation as well (R = 0.3224 (95% confidence interval: [0.0422 0.5557]), p = 0.03). On the other hand, when data from different emotional conditions were considered separately, none of the separate correlations between theta coherence at 800–1300 ms and habenula theta power at 2700–3300 ms were significant: (R = 0.3405 (95% confidential interval: [-0.1867 0.7154]), p = 0.2020 for Neutral; R = 0.3846 (95% confidential interval: [-0.1373 0.7394]), p = 0.1474 for Positive; R = -0.1655 (95% confidential interval: [-0.6111 0.3597]), p = 0.5474 for Negative). In addition, we tested whether this coherence-power correlation was specific to the time window identified based on Figure 6A. To do so, we quantified the correlation between the habenula theta power at 2700–3300 ms and the habenula-PFC theta coherence at -200–300 ms, 300–800 ms, 1300–1800 ms, and 1800–2300ms separately. None of the habenula-PFC coherences at other time windows correlated with habenula theta at 2700–3300 ms. We acknowledge that the effect shown in Figure 6C is weak and would not survive correction for multiple comparison. However, the selection of time window for the test shown in Figure 6C was based on the previous test shown in Figure 6A, not based on multiple tests.’